physiology

cobia, aerobic scope, critical oxygen concentration, exercise recovery, respirometry, Chesapeake Bay

**Author for correspondence:**
Daniel P. Crear
e-mail: dcrear8@gmail.com

# In the face of climate change and exhaustive exercise: the physiological response of an important recreational fish species

Daniel P. Crear[1], Rich W. Brill[1], Lauren M. L. Averilla[2], Sara C. Meakem[2] and Kevin C. Weng[1]

[1]Virginia Institute of Marine Science, William & Mary, Gloucester Point, VA 23062, USA
[2]William & Mary, Williamsburg, VA, 23185, USA

DPC, 0000-0002-9045-3649; KCW, 0000-0002-7069-7152

Cobia (*Rachycentron canadum*) support recreational fisheries along the US mid- and south-Atlantic states and have been recently subjected to increased fishing effort, primarily during their spawning season in coastal habitats where increasing temperatures and expanding hypoxic zones are occurring due to climate change. We therefore undertook a study to quantify the physiological abilities of cobia to withstand increases in temperature and hypoxia, including their ability to recover from exhaustive exercise. Respirometry was conducted on cobia from Chesapeake Bay to determine aerobic scope, critical oxygen saturation, ventilation volume and the time to recover from exhaustive exercise under temperature and oxygen conditions projected to be more common in inshore areas by the middle and end of this century. Cobia physiologically tolerated predicted mid- and end-of-century temperatures (28–32°C) and oxygen concentrations as low as 1.7–2.4 mg l$^{-1}$. Our results indicated cobia can withstand environmental fluctuations that occur in coastal habitats and the broad environmental conditions their prey items can tolerate. However, at these high temperatures, some cobia did suffer post-exercise mortality. It appears cobia will be able to withstand near-future climate impacts in coastal habitats like Chesapeake Bay, but as conditions worsen, catch-and-release fishing may result in higher mortality than under present conditions.

# 1. Introduction

Cobia (*Rachycentron canadum*) have become an important recreational fish, popular along the US mid- and south-Atlantic states [1]. With over 225 000 trips per year targeting cobia in Virginia alone and anglers valuing cobia fishing between $488 and $685 per trip, this recreational fishery is important for US coastal states [2]. Cobia are particularly vulnerable and heavily fished when they migrate into high salinity bays and estuaries, such as Chesapeake Bay, to spawn between May and September [3–5]. In recent years, landings of cobia along the US East Coast have exceeded the annual catch limit by over 200% [6]. This has led to the early or complete closure of the fishery in federal waters [7,8]. Although the cobia fishery remained open in most Atlantic states' waters, the controversy surrounding cobia management still persists.

To further complicate the situation, conditions in important coastal spawning habitats like Chesapeake Bay are expected to warm as a result of climate change [9]. The most common temperatures throughout the entire water column within Chesapeake Bay during the summer spans 24–28°C. With climate change, the water temperatures in Chesapeake Bay are expected to increase between 1.5 and 2°C by the mid-century and by 5°C by the end of century relative to the mid-1990s [10]. Therefore, as temperatures warm, we expect 28°C to occur even more frequently by mid-century and 32°C water temperatures to be very common by the end of century.

Due to a combination of warming and changes in precipitation through climate change and eutrophication through nutrient loading from human activities [9,11], we expect a decrease in dissolved oxygen within Chesapeake Bay. Specifically we predict to see the largest increase in cumulative hypoxic volume between oxygen concentrations of 2 and 5 mg l$^{-1}$ [12], which is too hypoxic for many marine species to inhabit [13]. This flux of hypoxic water that typically occurs mid-summer is expected to occur earlier in the year as well [12].

In a warming and increasingly hypoxic Chesapeake Bay, fish species intolerant of these conditions may have to shift their habitat selection. Theses shifts can result in a spatial mismatch between fish and their prey [14]. This becomes a particular problem for fish species [15] that prey on benthic invertebrates, which can often tolerate lower oxygen levels than many fishes [14,16]. Although cobia use the entire water column, two of their most common prey items, blue crabs (*Callinectes sapidus*) and Atlantic croaker (*Micropogonias undulates*) use benthic habitats [17]; therefore, if cobia are not as hypoxia tolerant as their prey, their diet may have to change if distribution shifts are inevitable in these coastal habitats.

A phenomenon that occurs as a result of warming and increasing hypoxia is called the 'temperature-oxygen squeeze' [18], where hypoxic bottom waters force fish to seek out more oxygenated, but warmer, surface waters. As a result, fish can be exposed to physiologically stressful temperatures or dissolved oxygen levels [14,18,19]. Further, the temperature-oxygen squeeze has forced species like striped bass (*Morone saxatilis*) in Chesapeake Bay to use warmer surface waters, which has increased their susceptibility to diseases, like *Mycobacterium* [19]. If cobia are negatively impacted by changing conditions, we may observe shifts in species distribution, which can lead to changes in migration [20,21], reproductive patterns [22] and abundance levels [22,23], ultimately influencing the population-level success.

We contend (as have others, e.g. [24–26]) that predicting the potential impacts of climate change on marine fishes requires that physiological abilities and environmental tolerances be quantified. If species are able to physiologically tolerate these changes, they could be considered a winner in the climate crisis. These physiological tolerances can be assessed through various measurements, including aerobic scope (AS), which is generally defined as the difference between maximum and standard metabolic rates, and within which all of life's processes must occur [27–29]. It appears that the trend of aerobic scope over temperature varies considerably on both a species-specific [26] and individual-specific level [30], where some (species or individuals) may display a bell-shaped curve [28,31], while others show a steady increase in AS with temperature up to the species' lethal limits [26,28]. Cobia are known to frequent subtropical waters; therefore, we expect cobia to have the capability to withstand warming waters over the next few decades. However, as conditions exponentially intensify by the end of century, it is unclear whether they will be able to withstand those changes. Hypoxia tolerance can be quantified through the measurement of an individual's critical oxygen saturation ($S_{crit}$), which is the oxygen level at which the individual can no longer maintain standard metabolic rate (SMR) [32]. Further, as temperature increases, hypoxia tolerance decreases (i.e. $S_{crit}$ increases) [33,34]. Due to cobia's exposure to low oxygen when they use bays and estuaries in the warm months, they are expected to be relatively hypoxia tolerant. Lastly, increasing ventilation volume is another compensatory mechanism that fish might employ to maintain oxygen delivery to the tissues and

overcome oxygen debt [35,36] and can thus be used like metabolism to indicate when environmental conditions stray from optimal [27,37]. We predict that like other species, cobia ventilation volume will track with their metabolic rate.

Cobia already have to endure capture, handling and air exposure when caught and will have to face environmentally stressful conditions when released in the future. Being released into conditions that do not promote recovery (e.g. low oxygen and non-optimal temperatures) could prolong recovery and reduce their chances of survival post-release. Recovery time post-exercise (angling) at various environmental conditions can be assessed using metabolic rate [38]. This type of information can help managers understand how well cobia may recover from capture post-release under the increased temperatures expected to occur in coastal habitats. As conditions become more stressful (i.e. warmer), we expect recovery time to increase. However, recreational catch-and-released data [39] show multiple cobia caught on subsequent days. This suggests stress levels have gone down enough where feeding is desired [40]; therefore, cobia may have a fast recovery time.

## 2. Objectives

A heavily fished population of cobia may be vulnerable to climate change because they rely on a habitat for feeding and spawning that is expected to worsen as a result of climate change. However, their tolerance to the potential changes in temperature and oxygen levels is unknown. Based on this, the objective of this study was to measure aerobic scope, $S_{crit}$, ventilation volume and exercise recovery time in cobia exposed to warm temperature and hypoxic waters similar to what we may expect to see in Chesapeake Bay as climate change persists through mid- and end of century. This information will help us understand how cobia may be impacted by climate change and whether they are a climate change winner, and in turn inform future management for this species.

## 3. Material and methods

### 3.1. Cobia collection and maintenance

Cobia (81–108 cm total length; 3.1–9.8 kg) were caught within Chesapeake Bay and brought back to the VIMS Seawater Research Laboratory (SRL). Individuals were held in a large swimming pool (6.7 m diameter; 32 000 l) with a recirculating system where mechanical and bio-filtration and periodic water changes were used to maintain water quality. Fish were acclimated to captivity for two weeks prior to experimentation and fed frozen menhaden (*Brevoortia tyrannus*) twice a week. All animal capture, maintenance and experiment protocols were approved by the College of William & Mary Institutional Animal Care and Use Committee (protocol no. IACUC-2017-05-26-12133-kcweng).

### 3.2. Experimental design

Maximum metabolic rate (MMR), SMR and $S_{crit}$ were measured at 24°C (current treatment), 28°C (mid-century treatment) and 32°C (end-of-century treatment) (sample sizes of 13, 11 and 10, respectively), using intermittent-flow respirometry [19,32]. Experimental temperatures were selected to represent commonly occurring summertime temperatures in Chesapeake Bay (24 and 28°C, with the latter expected to occur more often by mid-century) and a temperature (32°C) that rarely occurs today but that is expected to occur more often by end of century [10]. Cobia were acclimated to treatment temperatures in the holding tank for at least two weeks prior to experimentation. Because of mortalities suffered in the holding tank between experimental treatments, no individuals were tested at all three temperature treatments and nine were tested at two temperatures. Of the 13 cobia tested at 24°C, one was also tested at 28°C. Eight individuals were tested at both 28 and 32°C. The issue of using the same individual for multiple treatments is addressed in the Data analysis section below.

The respirometer system was composed of a rectangular plexiglass respirometer (110 or 214 l volumes based on fish size) submerged in a larger water reservoir. Oxygen in the reservoir was regulated by bubbling air (normoxia) or nitrogen gas (hypoxia), while temperature was set and controlled by a heat exchanger (Aqua Logic, San Diego, CA, USA). Oxygen levels in the respirometer and the reservoir were measured by two fibreoptic probes and a two-channel FireSting oxygen meter (Pyroscience, Aachen, Germany). One oxygen fibreoptic probe was fit through a stopper plugged into the top of the respirometer, whereas the reservoir's oxygen sensor was fixed along the side of the reservoir. Oxygen

levels in the respirometer and reservoir were calculated and displayed at one second intervals using FireStingO2 software. These data were subsequently relayed to a custom software developed and designed in Dasylab 9.02 (National Instruments, Austin, TX, USA), which controlled the pump supplying water to the respirometer (described below) and a solenoid valve controlling the flow of nitrogen gas to the outer bath (and thus regulating ambient oxygen levels). The software also recorded the temperature and oxygen data, and called an Excel Macro that calculated metabolic rate in real time [19,32]. A recirculating loop was used to ensure that water in the respirometer was mixed.

During a trial, the respirometry system cycled through two periods termed 'flushing' and 'measurement'. During the flushing period (7–15 min), oxygen- and temperature-controlled water was pumped from the reservoir into the respirometer and out through the chimney at the top of the respirometer. During the measurement period (4–7 min), the pump supplying water to the respirometer was turned off, allowing the cobia to reduce the oxygen level in the respirometer. The measurement period consisted of a 1–2 min equilibration interval to ensure oxygen was mixed in the respirometer, followed by 3–5 min of data recording for metabolic rate calculation. The slope of a linear regression model fitted to the oxygen measurements was used to calculate metabolic rate using the equation:

$$MO_2 = b \times V \times W^{-1},$$

where $MO_2$ = metabolic rate (mg $O_2$ $kg^{-1}$ $h^{-1}$), $b$ = rate of change of oxygen content (estimated slope of linear regression) over the 3–5 min recording period ($s^{-1}$), $V$ = respirometer volume (l) corrected for the volume of the fish and $W$ = weight of the fish (kg). Metabolic rate measurements where the regression $R^2$ value was below 0.8 were eliminated and assumed to be compromised due to either poorly mixed water in the respirometer or contact between the cobia and oxygen probe (which did occur rarely).

Microbial respiration was accounted for by measuring oxygen consumption without the presence of an experimental animal for at least 3 h before and after each trial [30,41]. A linear regression was used to estimate the rate of oxygen decline due to microbial respiration during the trial based on oxygen consumption before and after the trial. The rate of oxygen decline due to microbial respiration was subtracted from the rate of oxygen decline when the cobia was present [32,42].

## 3.3. Ventilation

Ventilation rate and mouth gape were recorded by a camera placed inside the reservoir (but outside the respirometer) that was directly facing the fish. Ventilation rate was defined as the number of buccal movements in 1 min during a measurement period. If the fish was not facing the camera for at least one minute during a measurement period, no ventilation rate was recorded for that measurement period. Interorbital width (when fish was facing straight on) and eye diameter (when side of fish was in view) were measured prior to the start of a trial and used as reference points to determine gape size. During the first ventilation of a measurement period, an image was isolated when the mouth was opened the widest. If the fish was not in a position where the first ventilation mouth gape was clear, an image of the second ventilation was taken and so on. If mouth gape was not visible during any time of the measurement period, mouth gape was not recorded for that measurement period. The isolated images were imported into ImageJ, where the reference point was selected (based on the position of the fish), measured and calibrated. After calibration, mouth gape size was determined as the distance from the tip of the snout to the lower jaw [35]. This distance was measured three times and the mean was used to represent mouth gape for a given measurement period. We assumed that mouth gape was proportional to tidal volume and that the buccal floor dropped with mouth gape. A metric which we considered to be representative of the volume of water inhaled during a respiratory cycle was created by combining ventilation rate and mouth gape. Within a trial, all mouth gape values were divided by the minimum mouth gape so that all values were relative to when the cobia was most calm. Each modified mouth gape value was multiplied by the corresponding ventilation rate. We assumed the resultant ventilation rate mouth gape metric (VRMG) to be proportional to ventilation volume.

## 3.4. Maximum and standard metabolic rates

After removal from the holding tank, an individual was transferred to a chase tank (3.0 m diameter, 2900 l volume) and given 30 min to recover from the transfer. We then used a chase protocol to induce burst swimming for 6 min or until the fish stopped responding [19,43]. Individuals were then subjected to a

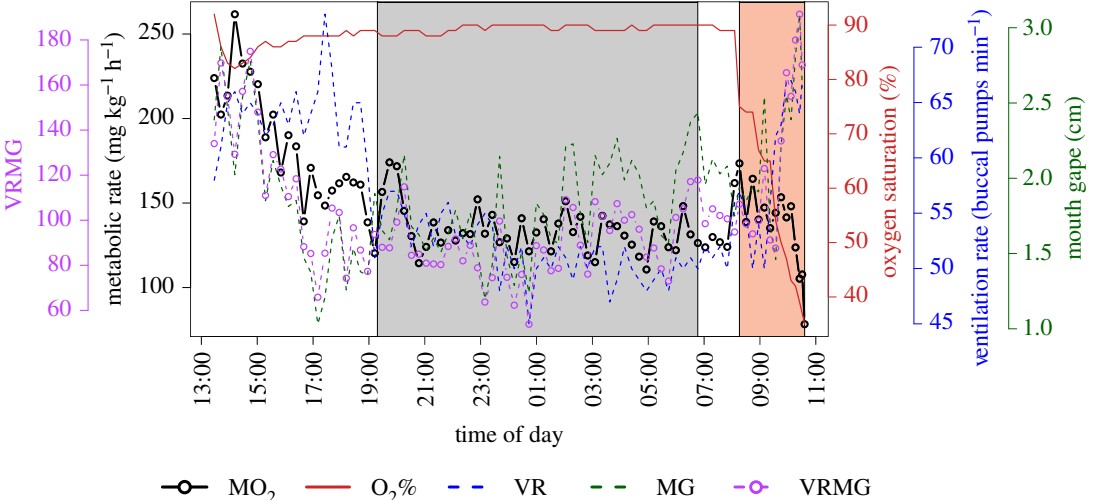

**Figure 1.** An example of a full metabolic rate trial at 32°C of a cobia (*R. canadum*). Each point corresponds to a calculated metabolic rate (MO₂) along with the associated oxygen saturation (O₂%), ventilation rate (VR), mouth gape (MG) and VRMG values. The grey-shaded area represents night-time. The red-shaded area represents the hypoxia period (O₂% < 80%). $S_{crit}$ for this individual at 32°C was 37% oxygen saturation.

1 min air exposure to ensure they were maximally exhausted so that MMR would be reached [28]. During air exposure, fish were weighed, and interorbital width and eye diameter were measured. Fish were then immediately transferred to the respirometer, which was then quickly sealed and the first metabolic rate measurement was initiated. Metabolic rate measurements were made for approximately 20 h to determine MMR and to allow the fish to recover from exhaustive exercise and reach SMR, which occurred once the metabolic rate levelled off and stabilized. This method of determining SMR was selected because the large size of our fish inhibited us from manipulating them too much and thus measuring a SMR baseline prior to the fish undergoing the chase protocol.

## 3.5. Critical oxygen saturation

Following the recovery period, the hypoxia part of the trial was initiated, where metabolic rates were measured as oxygen content was decreased in a stepwise fashion until the fish could no longer maintain its SMR. The oxygen level at which this decline in metabolic rate began to occur was considered $S_{crit}$ (calculated as described below). An example of a respirometry trial is shown in figure 1.

## 3.6. Recovery time

The time required for recovery from exercise was measured for each fish as the time elapsed between the fish being placed in the respirometer following the chase protocol and air exposure and the time when an individual's metabolic rate reached SMR [44]. This metric was assumed to represent the recovery time of a cobia after being hooked by a fisherman, pulled into the boat, air exposed and released back in the water.

## 3.7. Data analysis

MMR was taken as the highest metabolic rate within the first 3 hours of the trial. SMR was calculated from the mean of 10% of the lowest metabolic rate measurements (5–10 measurements) during the normoxic period (greater than 80% oxygen saturation). AS was calculated by taking the difference between MMR and SMR. In addition to calculating $S_{crit}$, $C_{crit}$ and $P_{crit}$ were also calculated, which are the concentration and partial pressure of oxygen, respectively, at which fish can no longer maintain SMR. $C_{crit}$ was taken as the first ambient oxygen value where an individual's metabolic rate dropped below SMR, and where the remaining metabolic rates were also below SMR. A linear regression was applied to those metabolic rates and the oxygen content associated with each measurement (2–11 measurements). The oxygen content where the regression line and SMR intersected was defined as $C_{crit}$ [30,34]. $C_{crit}$ values were unable to be calculated accurately using the above method for two individuals and for another fish we did not bring the oxygen content low enough in order to estimate

$C_{crit}$, so we only considered MMR, SMR and AS for those three individuals in the below analyses. $S_{crit}$ was calculated by dividing $C_{crit}$ by the oxygen concentration at 100% oxygen saturation at the treatment temperature. $P_{crit}$ was calculated by computing the partial pressure of oxygen at 1 atmosphere for the start day of the trial in Yorktown, VA, USA and multiplying by $S_{crit}$. MMR, SMR and AS of cobia that suffered a mortality during the normoxia part of the trial were consistently lower compared to those that did not die; therefore, those individuals that suffered a mortality were not included in any further analyses.

We developed a multivariate repeated measures model in SAS 9.4 software (SAS Institute) [45] using the MIXED procedure to examine the effect of temperature on MMR, SMR, AS and $P_{crit}$ similar to that described by Crear *et al.* [30] and Lapointe *et al.* [19]. The responses were MMR, SMR, AS and $P_{crit}$ for each trial and the covariate was temperature. The four responses were multiplied by a constant so that they were all on the same scale in order to maintain the assumption of normality. A random effect of individual was initially included in the model to account for using the same individual for multiple treatments, but the model did not change; therefore, to ensure parsimony, the random effect was not used in the final model. The Kenwood–Roger method was specified for calculating the degrees of freedom and the selected correlation structure was compound symmetry [46]. To quantify the effects of temperature on the four responses, *a priori* contrast statements of least-square means were made using the LSM estimate statement in SAS software. Model estimates were converted back to scale and all statistics were evaluated at a significance level of $\alpha = 0.05$. Predicted $P_{crit}$ and associated uncertainty was converted back to $C_{crit}$ and $S_{crit}$.

To determine how ventilation was related to metabolic rate, we fitted linear mixed effects models to data from individuals where ventilation rate and mouth gape were measured. Metabolic rate measurements were not included in these models when there were no associated ventilation rate or mouth gape available. Separate models were applied to data collected during the normoxic (oxygen saturation $\geq 80\%$) and hypoxic parts (oxygen saturation less than 80%) of the trials. The covariates considered for the normoxic model were VRMG, temperature, total length and the time since completion of the chase protocol. Due to the high amount of variability in VRMG among individuals, VRMG values were normalized to the maximum value within the trial. The random effect for the model was individual. We assessed the full model for heterogeneity and correlation structure. Based on Bayesian information criterion (BIC), modelling heterogeneity in responses among temperature treatments and using the autoregressive of order 1 (AR1) correlation structure was supported. In order to meet the assumptions of equal variances and normality, the metabolic rate was log transformed. A series of models were then developed using biologically meaningful combinations of covariates listed above, and the model with the most support was selected using BIC. The covariates considered for the hypoxic model were normalized VRMG, oxygen content, temperature and total length. Like the normoxic model, heterogeneity in responses among temperature treatments and correlation structure was supported on the full hypoxic model based on model selection. Multiple models were developed using the different combinations of covariates mentioned above and the model with the most support was selected based on BIC. Estimated marginal means were used to predict metabolic rates for the supported normoxic and hypoxic models [47]. Uncertainty estimates were generated using 1000 bootstrapped samples [48]. All linear mixed effects models were developed using the nlme package in R v. 3.4.3 [49].

To determine how temperature impacted recovery time, a model selection approach similar to above was used where potential covariates were temperature, AS and total length, and the random effect was individual. The full model was assessed for heterogeneity, correlation structure and a random effect, but those attributes were not selected in the most supported model based on BIC. Multiple linear models were then assessed based on various combinations of covariates and the most supported model was selected.

# 4. Results

## 4.1. MMR, SMR, AS and critical oxygen level

The evaluation of model diagnostics (plots of residuals) indicated that the multivariate repeated measures model provided acceptable fit to the raw metabolic rate metrics. Referenced model response variables (e.g. SMR, AS and metabolic rate) below represent model estimates, while raw values are indicated as such. Model results indicate significant differences for all four metrics among temperatures, primarily between the end-century treatment (32°C) and both the current (24°C) and

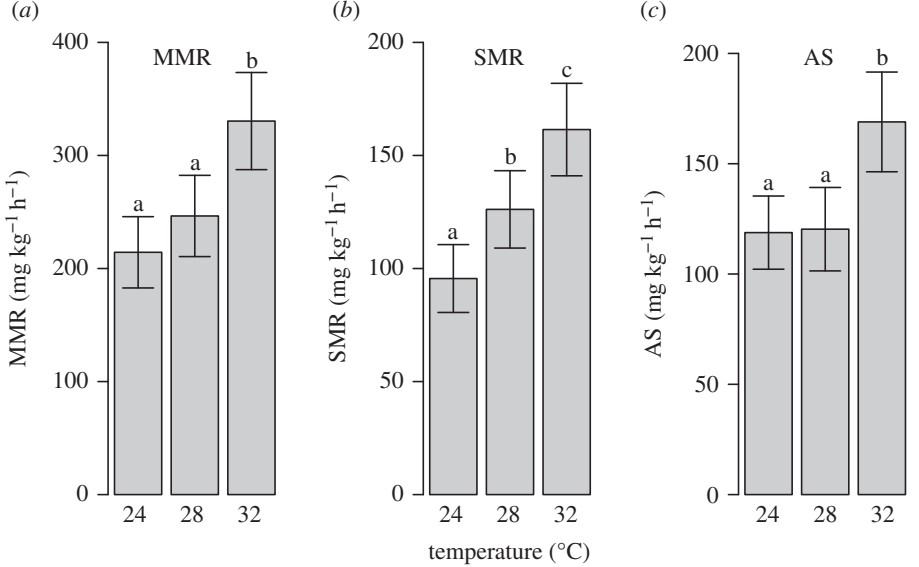

**Figure 2.** (*a*) MMR, (*b*) SMR and (*c*) AS at 24, 28 and 32°C of cobia (*R. canadum*). Data are model estimates of the mean ± 95% CI. Different lower case letters indicate a significant difference between two temperatures.

**Table 1.** T statistics from the *a priori* contrast statements of least-square means generated from the multivariate repeated measures model to examine the impact of temperature (24, 28 and 32°C) on MMR, SMR, AS and $P_{crit}$ of cobia (*R. canadum*). The degrees of freedom are in parentheses. The (*) signifies a significant difference between the two temperatures. The percentage value represents the per cent change between the second temperature (e.g. 28) and the first temperature (e.g. 24).

| metric | 24 × 28 | 24 × 32 | 28 × 32 |
|---|---|---|---|
| MMR | $t_{(91)} = -1.34$ (15%) | $t_{(91)} = -4.33^*$ (54%) | $t_{(91)} = -2.98^*$ (34%) |
| SMR | $t_{(91)} = -2.67^*$ (32%) | $t_{(91)} = -5.16^*$ (69%) | $t_{(91)} = -2.63^*$ (28%) |
| AS | $t_{(91)} = -0.12$ (<1%) | $t_{(91)} = -3.55^*$ (42%) | $t_{(91)} = -3.28^*$ (40%) |
| $P_{crit}$ | $t_{(92.41)} = -0.11$ (<1%) | $t_{(92.14)} = -3.90^*$ (48%) | $t_{(92.14)} = -3.50^*$ (46%) |

mid-century (28°C) treatments (figure 2 and table 1). From current to end-of-century treatments, MMR significantly increased 54%, while from mid- to end-of-century treatments it significantly increased 34%, and from current to mid-century, it only increased 15% and did not significantly differ (figure 2*a* and table 1). Significant differences occurred in SMR among all three temperatures treatments, where SMR increased 32% by mid-century and 69% by end of century relative to the current temperature treatment (figure 2*b* and table 1). Although significant differences in AS did not occur between current and mid-century treatments, AS did increase significantly by at least 40%, when comparing current and mid-century treatments to the end-of-century treatment (figure 2*c* and table 1). Lastly, similar to MMR and AS, significant increases occurred in $P_{crit}$ between the current and end-of-century treatments (48%) and mid- and end-of-century treatments (46%) (figure 3*a* and table 1). Raw MMR, SMR, AS, $C_{crit}$ and $S_{crit}$ are displayed in the electronic supplementary material.

## 4.2. Metabolic rate and ventilation during normoxia

The normoxic model that included VRMG, temperature, total length and time since the end of the chase and air exposure protocol provided the most parsimonious support. Each of these covariates were important in explaining variation in cobia metabolic rate data. The conditional $r^2$, which is the proportion of variance explained by both fixed and random covariates, was 0.71, while the marginal $r^2$, which is the proportion of variance explained by only the fixed covariates, was 0.65. The log of the predicted metabolic rate increased $0.10 \pm 0.04$ mg $O_2$ kg$^{-1}$ h$^{-1}$ for every unit increase of normalized VRMG (figure 4), while for every hour after being chased, the log of the estimated metabolic rate decreased $2.2 \times 10^{-2} \pm 1.5 \times 10^{-3}$ mg $O_2$ kg$^{-1}$ h$^{-1}$. For every centimetre increase in cobia total length, the

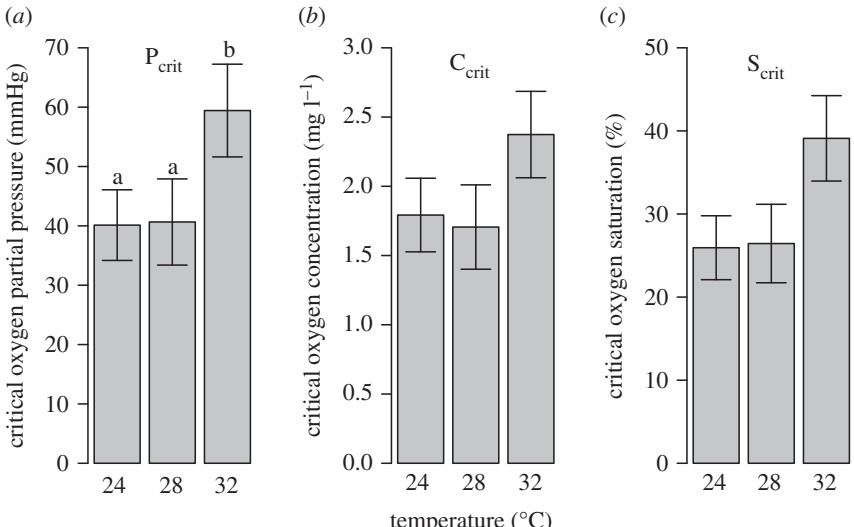

**Figure 3.** (a) Critical oxygen partial pressure ($P_{crit}$), (b) critical oxygen concentration ($C_{crit}$) and (c) critical oxygen saturation ($S_{crit}$) at 24, 28 and 32°C of cobia (*R. canadum*). $P_{crit}$ data are model estimates of the mean ± 95% CI. $C_{crit}$ and $S_{crit}$ were calculated from $P_{crit}$ model estimates. Different lower case letters over the $P_{crit}$ bars indicate a significant difference between two temperatures.

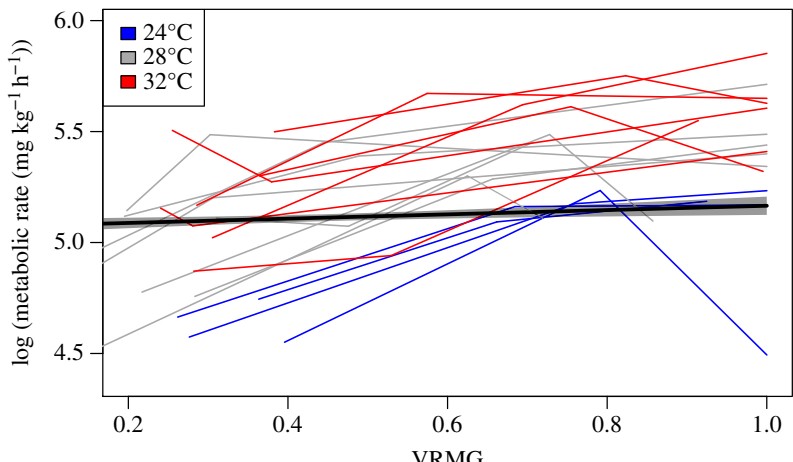

**Figure 4.** Log transformed metabolic rate values by normalized VRMG when cobia (*R. canadum*) are exposed to normoxic waters. The thick black line and the associated shaded region around it represent the log of the estimated metabolic rate values and the 95% CI, respectively. The coloured lines were generated from a segmented model for each individual trial to display how the log transformed metabolic rate values plateau at higher normalized VRMG values for a given temperature treatment. Segmented lines were only displayed for trials where there were 30% or more VRMG values relative to the overall number of measurement periods within the trial.

log of metabolic rate decreased $1.0 \times 10^{-2} \pm 2.3 \times 10^{-3}$ mg $O_2$ kg$^{-1}$ h$^{-1}$. The log of the predicted metabolic rate increased as temperature increased from 24°C ($4.9 \pm 0.3$ mg $O_2$ kg$^{-1}$ h$^{-1}$) to 28°C ($5.1 \pm 0.3$ mg $O_2$ kg$^{-1}$ h$^{-1}$) to 32°C ($5.3 \pm 0.3$ mg $O_2$ kg$^{-1}$ h$^{-1}$). Four cobia died during normoxia following chasing to exhaustion and air exposure; one at 28°C and three at 32°C. No cobia died during normoxia at 24°C.

## 4.3. Metabolic rate and ventilation during hypoxia

The hypoxic model that provided the most empirical support included temperature, total length and an interaction between VRMG and oxygen content. The conditional and marginal $r^2$ values were 0.79 and 0.71, respectively. For every centimetre increase in total length, predicted metabolic rate decreased $1.3 \pm 0.4$ mg $O_2$ kg$^{-1}$ h$^{-1}$. Estimated metabolic rate was $107 \pm 4$ mg $O_2$ kg$^{-1}$ h$^{-1}$ at 24°C, $138 \pm 5$ mg $O_2$ kg$^{-1}$ h$^{-1}$ at 28°C and $170 \pm 6$ mg $O_2$ kg$^{-1}$ h$^{-1}$ at 32°C. Lastly, as oxygen content increased, normalized VRMG increased slightly, along with predicted metabolic rate (figure 5). As oxygen content decreased,

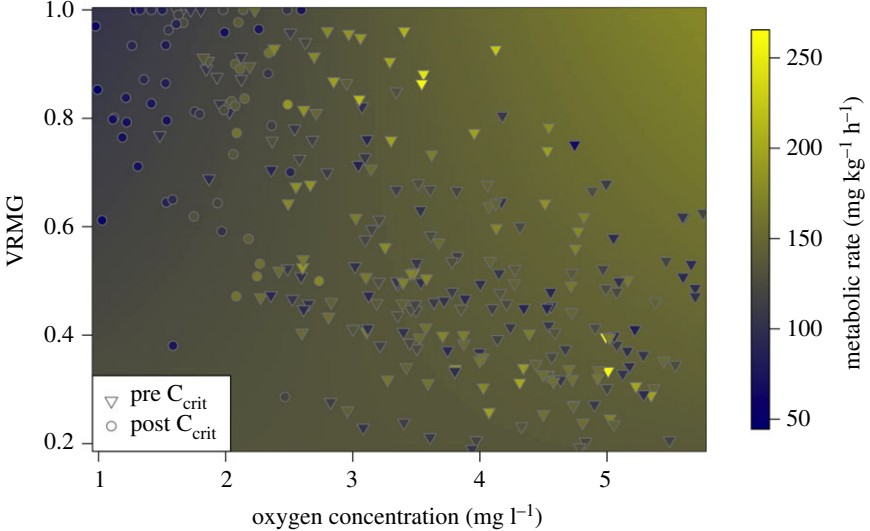

**Figure 5.** Cobia (*R. canadum*) metabolic rate in relation to normalized VRMG and oxygen content during hypoxia (less than 80% oxygen saturation). Symbols represent observed data, where inverted triangles are before $C_{crit}$ was reached and circles are after $C_{crit}$ was reached. The colours on the symbols represent the observed metabolic rate while the surface behind the symbols represents the predicted metabolic rate estimated from the model.

however, normalized VRMG increased whereas estimated metabolic rate decreased. This decrease in metabolic rate was particularly pronounced when normalized VRMG was higher and oxygen content was below $C_{crit}$ (1.7–2.4 mg l$^{-1}$ at 24–32°C).

## 4.4. Recovery time

The model that provided the most parsimonious support included an interaction between temperature and AS ($r^2 = 0.69$). We found that at the current temperature treatment (24°C), recovery time did not change with AS. At the mid-century treatment (28°C), however, recovery time increased by 6.0 min for every unit increase in AS, and at the end-of-century treatment (32°C), recovery time increased by 4.5 min for every unit increase in AS. Overall, recovery time was shorter at high temperatures.

# 5. Discussion

## 5.1. Effects of temperature under normoxia

Cobia are generally considered to be eurythermal, as they experience a wide range of temperatures (20–32°C) as adults when they spawn in estuaries like Chesapeake Bay during the summer months [1,3] or as juveniles in estuarine nursery areas. The tolerance of cobia to high temperatures is also supported by the absence of mortalities while individuals were held at 32°C for over a month during acclimation and experimentation periods.

Our results showed that AS of cobia continued to increase with temperature up until almost lethal temperature; however, when put under intense stress at high temperatures, it became lethal. This trend is in direct contrast to oxygen- and capacity-limited thermal tolerance (OCLTT) theory, which suggests that AS declines at temperatures above and below an optimal temperature because of reductions in the capacity of the ventilatory and circulatory systems to deliver oxygen to the tissues, resulting in a decline of AS [28,31,50]. According to the OCLTT theory, AS should therefore display a bell-shaped relationship with temperature. A number of studies have, however, found that the AS of fishes continues to increase up to lethal limits [30,51–53], as it does in cobia. During these studies, SMR increased with temperature (as expected) due to the impact of temperature on chemical and biochemical reactions [25,54]. MMR did not, however, decline or level off as temperatures approached lethal limits, leading to an increase in AS up to a knife-edged metabolic collapse. For example, in barramundi (*Lates calcarifer*) AS increased with temperature and was maximized at 38°C, but when given the option, barramundi only selected temperatures greater than 36°C 10% of the time [50]. Pink

salmon (*Oncorhynchus gorbuscha*) AS was maximized at 21°C, but between 17 and 21°C cardiovascular function showed signs of impairment [53]. Although cobia AS was maximized at 32°C, three out of 10 individuals died at 32°C within 3.7–7.8 h post-exercise suggesting cobia can incur irreversible physiological damage at this temperature after being exercised to exhaustion. Without further studies on the physiology of cobia, such as quantifying the effect of temperature on blood oxygen binding, gill gas exchange rates and heart function, it is difficult to explain exactly what is causing the patterns in AS we observed.

VRMG closely tracked metabolic rate under normoxia. However, at the highest VRMG values within a trial, this relation did not hold and metabolic rate values plateaued instead of continuing to increase (figure 4). This pattern is highlighted by the segmented lines in figure 4 and occurred in 79% of trials that had enough VRMG values (i.e. trials with VRMG data recorded for greater than 30% of the total number of measurement periods within the trial). This may occur because oxygen utilization (i.e. the fraction of oxygen removed from the ventilatory water stream) decreases as ventilation volume increases due to an increase in physiological dead space or an increasing ventilation : perfusion mismatch [55].

## 5.2. Effects of hypoxia

Overall, cobia appear to be hypoxia (less than 2.0 mg l$^{-1}$) tolerant. Their $C_{crit}$ and $S_{crit}$ values are below 2.0 mg l$^{-1}$ and 30% oxygen saturation, respectively, at temperatures (24 and 28°C) they currently experience while occupying mid-Atlantic estuaries and that they are likely to encounter more often by mid-century [10]. Other active migratory species that use Chesapeake Bay, like sandbar sharks (*Carcharhinus plumbeus*) and striped bass (*Morone saxatilis*), have much higher $C_{crit}$ (3.3 ± 0.2 mg l$^{-1}$; 2.5 ± 0.2 mg l$^{-1}$, respectively) and $S_{crit}$ values (51 ± 2%; 35 ± 2%, respectively) at 28°C [19,30]. Cobia $P_{crit}$ values (40 mmHg at 24 and 28°C) were as low as less active Chesapeake Bay species such as the summer flounder (*Paralichthys dentatus*) (42 mmHg; [56]) at 22°C. When they enter estuaries like Chesapeake Bay they may be required to enter hypoxic zones to hunt for specific prey items. For example, common prey items for cobia [17] like blue crabs and Atlantic croaker have $C_{crit}$ values of 1.6–2.5 mg l$^{-1}$ (23–28°C; [32]) and 1.8 mg l$^{-1}$ (25–30°C; [57]), respectively. Although hypoxia tolerance of cobia significantly worsened at the end-of-century treatment (32°C), it still did not exceed 2.5 mg l$^{-1}$.

Unlike during normoxia, VRMG did not track metabolic rate as closely when the fish were in hypoxia (figures 1 and 5). As oxygen content dropped, fish were forced to increase their ventilation volume to compensate for the lower oxygen content of the ventilatory water stream. However, similar to when cobia were exercised, as ventilation volume increased oxygen utilization decreased. This effect, combined with a decreased water oxygen content, led to the observed decoupling of VRMG and metabolic rate. Oxygen demand increases with temperature even in hypoxia, implying that during hypoxia a negative feedback loop is created. Although cobia are likely to avoid areas with this combination of stressful conditions in Chesapeake Bay (i.e. elevated temperatures and reduced oxygen levels), avoidance may become more difficult as conditions worsen within Chesapeake Bay during the summer months due to the effects of climate change, particularly by end of century. Due to the importance of Chesapeake Bay to cobia spawning, fish may have to endure less suitable conditions to ensure spawning occurs.

## 5.3. Recovery post-exercise

Typically as AS increases, recovery time following exhaustive exercise is shorter [44], but we actual found the opposite for the warmest temperatures. In addition, recovery time was shorter at high temperatures when AS was higher (for 32°C). A quicker recovery time with temperature may be attributed to a faster lactate breakdown in warmer temperatures [58]; however, mortalities did occur post-exercise at higher temperatures. Therefore, it is likely that at higher temperatures there are physiological effects which are not reflected by changes in AS. These trends are difficult to explain without measuring various physiological parameters such as white muscle and blood non-bicarbonate buffering capacity, and post-exercise blood and muscle lactate and glycogen levels [58–60]. Lastly, it is important to note that by measuring SMR (baseline) after the fish appeared to be recovered from the chase protocol instead of getting a baseline first, we may have overestimated SMR and underestimated recovery time.

## 5.4. Impacts of climate change

Climate change is expected to negatively impact many marine species [61–63]. However, it appears that cobia may be more resilient to the changes (i.e. elevated temperatures and reduced oxygen levels)

expected to occur in Chesapeake Bay by mid-century [10]; therefore, we believe they will be a climate change winner at least in the next few decades. Alternatively, if conditions become more favourable earlier in the year or in more northward locations, cobia may move into Chesapeake Bay earlier in the year or more frequently occupy coastal waters north of Virginia. Their adaptations to the variable environmental conditions in coastal habitats appear to allow cobia to withstand high temperatures and low ambient oxygen levels. The lack of mortality or change in behaviour while in the holding tank for over a month at the end-of-century treatment suggests that chronic exposure to high temperatures may not be detrimental to this species.

## 5.5. Impacts of fishing

Despite withstanding end-of-century conditions, it is important to note that if exercised to exhaustion their risk of mortality is elevated at high temperatures and low ambient oxygen levels. In other words, as environmental conditions change, cobia may more frequently occupy areas with elevated temperatures and low ambient oxygen levels, and rates of post-release mortality are likely to increase. Further, when stressed under high temperatures, the potential to contract a disease or parasite increases, which can negatively impact metabolism [19,64].

The potential for cobia to undergo stressful events has only increased as they have seen a rise in fishing pressure over recent years. Increasing fishing effort and stricter regulations have most likely increased the number of cobia discarded in the recreational fishery. Therefore, in addition to being overfished due to direct fisheries mortality, cobia will be at a higher risk of post-release mortality as inshore areas warm and hypoxia increases in extent and severity [9,10,12]. With increased fishing effort and regulations resulting in higher rates of catch-and-release fishing, more individual cobia will be forced to recover ($13.7 \pm 0.9$ h) from exhaustive exercise and air exposure in a habitat that is getting warmer and less oxygenated due to climate change, potentially resulting in post-release mortality and also impacting predator avoidance, feeding ability and spawning success. Further, we may see greater detrimental effects on species that lack cobia's high tolerance to more extreme conditions, but endure similar catch-and-release fishing.

To improve future post-release survival as conditions worsen, fisheries management could encourage leaving the fish in the water post capture prior to release. Currently all cobia caught are typically brought into the boat, dehooked and are either kept or released. Regulations have been put into place to leave some important recreational fish species (e.g. white marlin [65]) in the water. Further, by limiting the fishing season to months that are less warm (e.g. May, June and September), we would avoid releasing stressed fish into suboptimal conditions. Although these changes are not likely to be warranted for today's conditions, we do believe they may reduce mortality in the coming decades.

## 6. Conclusion

We identified cobia's response to predicted conditions expected to occur in Chesapeake Bay. We found that cobia can withstand temperatures as high as 32°C. Cobia appear to be a climate change winner in the near future (30+ years) due to their physiological abilities to survive environmental conditions predicted to occur by mid-century in Chesapeake Bay and elsewhere in western Atlantic coastal areas. The hypoxia tolerance of cobia (i.e. relatively low $C_{crit}$) may allow this species to follow prey items in hypoxic regions of their coastal habitats. Further, being a generalist feeder and selecting prey items that also are hypoxia tolerant will benefit cobia in the future when their prey are still present despite worsening conditions. However, compounding stressors caused by catch-and-release recreational fishing in areas with elevated temperatures could cause cobia populations to be more sensitive to the effects of climate change, particularly when conditions reach end-of-century levels within estuaries like Chesapeake Bay. This type of information can be useful to future managers who will have to consider the interactions among fish species, the changing environment and various fishing practices.

Data accessibility. Data available from the Dryad Digital Repository: https://doi.org/10.5061/dryad.zgmsbcc61 [66].
Authors' contribution. D.P.C. substantially contributed to project conception and design, animal capture, collection of all physiology data, analysis and interpretation of data, and drafting the manuscript; R.W.B. participated in the design of the study, helped interpret physiology data and revised the manuscript; L.M.L.A. substantially participated in animal care, ventilation data collection and analysis and helped revise the manuscript; S.C.M. significantly helped

with animal collection and care, the ventilation data collection and analysis, conducting respirometry experiments and contributed to manuscript revisions; K.C.W. contributed funds to the project, participated in project conception and design, animal collection and assisted in manuscript revision. All authors gave final approval for publication.

Competing interests. The authors have no competing interests.

Funding. This work was supported by the Virginia Institute of Marine Science, College of William & Mary startup funds and Virginia Sea Grant.

Acknowledgements. We would like to thank William & Mary graduate and undergraduate students, including S. Kilmer, C. Corrick, B. Gallagher and J. Buchanan, for their help catching and bringing cobia back from Chesapeake Bay. We thank S. Kilmer and G. Schwieterman for their help setting up many of the respirometry trials. We particularly thank Brian Watkins along with many other fishermen for catching cobia that we could bring back to VIMS. We also would like to thank Mary Fabrizio for assistance in statistical analysis.

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
