## [Reviewer comments · Royal Society Open Science]

Review History

RSOS-191704.R0 (Original submission)

Review form: Reviewer 1

Is the manuscript scientifically sound in its present form?

No

Are the interpretations and conclusions justified by the results?

Yes

Is the language acceptable?

Yes

Do you have any ethical concerns with this paper?

No

Have you any concerns about statistical analyses in this paper?

Yes

Recommendation?

Reject

Comments to the Author(s)

This study sought to address the vulnerability of an economically important fish species to global climate change scenarios through a physiological lens. Without a doubt, understanding species-specific responses to climate change is important, especially in threatened or economically beneficial species, but I do not think the authors were able to successfully support their claims in this submission. My comments are as follows:

Introduction:

The introduction struggles to find a balance between discussing the consequences of stress from recreational (angling) and climate change (temperature and hypoxia) sources.

Lines 98-99: A citation is needed to provide evidence of this relationship.

Lines 100-105: I think this paragraph needs to be reworked. Why is understanding how recovery time changes important and what would that mean? Why the emphasis on angling?

The hypotheses and predictions should be stated more clearly in the introduction. The authors are measuring a large number of traits, and it would be best to clearly lay out expectations to the reader.

Methods:

Lines 133-134: The authors note that there were unexpected mortalities and some individuals were exposed to multiple temperature treatments. It is unclear if this was corrected/controlled for in their analyses. From the data provided, it appears at least 9 individuals were measured under multiple temperature treatments, and if order of temperature exposure is important, then this could cause major differences in their results.

The measurements of metabolic traits raised concern. The order of measurements (maximum, post exercise, standard, and critical oxygen tension) could potentially influence the interpretation of the results. By beginning with measuring maximum metabolic rate, the authors did not acclimate the fish to the respiratory chamber or establish a relative baseline value that would be considered standard metabolic rate. Without having the standard metabolic rate baseline prior to measuring recovery, it is possible the authors are actually measuring the return to baseline values.

Additionally, the order of experiments could influence the outcomes. If possible, it would be best to know if the authors conducted preliminary tests to know whether order of operations influenced the results of the experiment.

Data analysis:

Mass was not included in the models tested, but it is known to be tightly correlated with metabolic traits.

Results:

Lines 286-299: The authors note the percent differences between current to end-of-century and current to mid-century comparisons. These percent differences are interesting, but could potentially be misleading to readers about the statistical significance of the findings. For instance, in lines 297-298, the authors state "Lastly, similar to MMR and AS, significant increases occurred in Pcrit between the current and end-of-century treatments (48%) and mid- and end- of century treatments (46%)." No p-values are provided, and the use of percentages here may create confusion as to whether or not the current and mid-century treatments are different.

Additionally, the authors seem to use mid-century to describe the 28 degree treatment, but in their introduction state that water temperatures in the Chesapeake Bay already span the 24-28 degree range. It is unclear to me how this does not also reflect current trends.

Discussion:

The discussion should be reworked to reflect changes and make grammatical corrections but is otherwise fine.

Figures:

Figure 1: Figure 1 is very complicated and is difficult to read. The information provided in this graph needs to be separated into multiple figures that could be included in the online supplement.

Figure 4: The gray lines are difficult to see. I think this could be included in the supplement as well.

Figure 5: This figure is difficult to read given the extent it relies on color for interpretation. It is not color-blind friendly. I think it could be simplified by creating multiple figures highlighting these key relationships.

Review form: Reviewer 2

Is the manuscript scientifically sound in its present form?

Yes

Are the interpretations and conclusions justified by the results?

Yes

Is the language acceptable?

Yes

Do you have any ethical concerns with this paper?

No

Have you any concerns about statistical analyses in this paper?

No

Recommendation?

Major revision is needed (please make suggestions in comments)

Comments to the Author(s)

GENERAL COMMENTS:

- Thank you for the opportunity to review manuscript RSOS-19704. This is a generally well-constructed draft and the paragraphs are mostly well-written with few typos which makes the review process significantly easier on a reviewer. Much appreciated.

- Though "climate change winner" is in the title - it only receives a cursory nod in the conclusion - it should be present throughout the text; introduced very early on, highlighted in the results, and interpreted in the discussion.

- Consideration of management implications is warranted as a new section. What are potential options for regulations to address anticipated impacts of post-release mortality?

- Title: smart title - should do well with search engines

- Abstract: clear and concise
- Keywords: note, for keywords, it is generally most useful to expand literature search processes by including words that are NOT in the title – so no need to state “climate change” but can add an additional relevant term, such as Chesapeake Bay
- Introduction: The content here could be much more effectively presented with extensive restructuring. I would recommend starting with importance of Chesapeake Bay and cobia, anticipated climate change (including a temperature-oxygen squeeze) within the Bay, anticipated impacts to cobia, and conclude with rationale for how to examine.
- Methods: Please clarify – were some individuals used for multiple trials? Making for non-independent samples?
- Results: See line-by-line note below about treatment labels – they are present in this section (and the abstract) but should be introduced much sooner in the text.
- Discussion: What about chronic impacts? I realize that is beyond the scope of this experiment but can you draw some interpretation based on the findings and other literature? I would also like to see more of the “climate change winner” language woven throughout this section. I also recommend breaking the “impacts of climate change and fishing” into two sections. I think doing so will allow for a more nuanced interpretation on “projected climate impacts” that will be a valuable addition to the piece.
- Conclusion: This section can “push the envelope” a bit further than just providing a summary of the results. It’s worth noting that this is the first time in the text that “climate change winner” is mentioned (though it is in the title). This conclusion could be more valuable in qualifying the “winner” status as well as providing some discussion on management recommendations for how to address anticipated impacts.
- Table 1: need to indicate the species and why these temperature points where selected.
- Figure 1: efficient use of y-axes but a bit more spacing/separation would make them more legible. Include species name in caption?
- Figure 2, 3, 4, 5: include scientific name in caption? Also rationale for temperature selections.
- Figure 2, 3: Could consider combining y-axes as in figure 1 to create group bar charts to more simply compare the metrics across each temperature point more simply?
- Figure 5: define Ccrit in caption.
- References: not reviewed

LINE-BY-LINE COMMENTS:

40: add “mid- and end-of-century temperatures”

54-60: This paragraph could be stronger – introduce cobia, why the fishery is important, and the issue of its current status.

57: add comma after “years”; landings “have”

61-71: I would divide this into two paragraphs – one on anticipated temperature change and one on anticipated oxygen concentrations. These are the two important premises of the piece and they should establish the foundation for the study.

62: “worsen” is value-laden – consider a more objective terminology – e.g., alter? Change? Warm?

72-84: This section should build from the prior two – complex ramifications of climate change – it’s quite cursory as it currently stands and doesn’t particularly relate to cobia

77-79: how relevant to cobia? Why include?

85-99: I think this should be introduced after setting up the concern with cobia

95: comma needed after “increases”

106: transition here is somewhat abrupt between topics – consider breaking this out into its own “objectives” section?

130: some on anticipated temperatures in the Bay in the introduction is warranted to ground the decision choices here.

133: clarify – some individuals were used for multiple trials? I.e., non-independent samples? What proportion of your trials? What is the implication of that for your results and interpretation?

151: period should be inside quotation marks

168: citation for 3h period to be sufficient for microbial respiration?

224: extra space before period

251: citation for SAS?

263: spell out BIC on first mention

264: include a brief mention of what AR1 correlation structure is

289: the temperature treatments aren’t introduced in the methods with the “end-of-,” “mid-century,” and “current monikers but they should be. Actually, even better would be to introduce them this way in the objectives.

341: evinced? Word choice?

343: if using “e.g.,” “etc.” is superfluous.

361: Can you extend this discussion to more chronic exposure? I realize that was not within the confines of this experiment but possibly by couching in additional literature?

363: I think this rationale is suspect, as evident by the subsequent qualification, to the point where it would be better not to include it as support for the prior statement. Unless there was some way to subset the records by when high temperatures were present but still, I think this is questionable at best.

368: but I think it goes a far way to show that the pattern is exhibited in other fishes as well.

379: define here the oxygen threshold for hypoxia in parenthetical

382: citation for climate projection?

395: reference figure?

406: take this statement further – avoidance behavior is somewhat a “catch-all” for negating projected climate impacts – elaborate on why this will be more difficult to contradict that argument.

408: This section is somewhat jumbled -- mixing interpretation of the experiment with climate projections. I think it would be clearer if it was revised into two distinct sections: 1) Post-release mortality effects and 2) projected climate impacts

412: This alternative hypothesis was not examined in this study – either reference additional research or just don't include as it wasn't a part of this work.

415-421: This is an important qualification for why cobia wouldn't actually end up a “climate change winner.” I think it would be more appropriate to break this out as its own paragraph.

422-427: This jumps to implications for the fishery – I think it would be better presented later in the discussion after wrapping up the physiological impacts.

428: any other examples in the literature (cobia or other species) that found similar results?

444: Did this experiment identify temperature and oxygen thresholds? Or rather thresholds at pre-determined temperature points that correlate with expected temperatures mid- and late-century?

449: I think further qualification of the “climate change winner” status is needed – what about prey abundance mid- and late-century? What fishing regulations could help cope with the changes?

Decision letter (RSOS-191704.R0)

02-Jan-2020

Dear Mr Crear:

Manuscript ID RSOS-191704 entitled "A climate change winner: Measuring the effects of warming and hypoxia on the metabolism of an important recreational fish species" which you submitted to Royal Society Open Science, has been reviewed. The comments from reviewers are included at the bottom of this letter.

In view of the criticisms of the reviewers, the manuscript has been rejected in its current form. However, a new manuscript may be submitted which takes into consideration these comments.

Please note that resubmitting your manuscript does not guarantee eventual acceptance, and that your resubmission will be subject to peer review before a decision is made.

Once you have revised your manuscript, go to <https://mc.manuscriptcentral.com/rsos> and login

to your Author Center. Click on "Manuscripts with Decisions," and then click on "Create a Resubmission" located next to the manuscript number. Then, follow the steps for resubmitting your manuscript.

Your resubmitted manuscript should be submitted by 01-Jul-2020. If you are unable to submit by this date please contact the Editorial Office.

on behalf of Dr Michael Tobler (Associate Editor) and Kevin Padian (Subject Editor)
openscience@royalsociety.org

Associate Editor Comments to Author (Dr Michael Tobler):

The manuscript has been assessed by two reviewers. Both found merit in the study, but also pointed out some substantial methodological issues. I agree with the reviewers that the multiple use of individual fish, mortality during trials, and order in which different aspects of metabolism were raise serious issues that need to be addressed. If the authors feel like they can address some of the major issues identified by the reviewers, the manuscript could be resubmitted.

I also have some additional suggestion on how the authors could improve the effectiveness of their figures:

- Figure 1: Avoid graphs that plot multiple dependent variables and have multiple y-axes. Instead, create multi-panel figure where each dependent variable is plotted on top of the other. That still allows the reader to detect trends across variables, but it's much easier to figure out what data points belong to what axis.
- Figures 2 and 3: Instead of plotting mean and CI, consider a form of data visualization that shows the data (or at least data distribution). See here for more information: <https://journals.plos.org/plosbiology/article?id=10.1371/journal.pbio.1002128>
- Figure 5: Please consider alternative color scales to make the figure easier to interpret: See here for more information: <https://journals.plos.org/plosone/article?id=10.1371/journal.pone.0199239>

Subject Editor comments:

Thanks for your submission. As you will see the referees like many things about it but have some fundamental concerns. Please be sure to address these specifically in your response letter, should you choose to resubmit, so that we are able to consider your resubmission further. Best wishes.

Reviewers' Comments to Author:

Reviewer: 1

Comments to the Author(s)

This study sought to address the vulnerability of an economically important fish species to global climate change scenarios through a physiological lens. Without a doubt, understanding species-specific responses to climate change is important, especially in threatened or economically beneficial species, but I do not think the authors were able to successfully support their claims in this submission. My comments are as follows:

Introduction:

The introduction struggles to find a balance between discussing the consequences of stress from recreational (angling) and climate change (temperature and hypoxia) sources.

Lines 98-99: A citation is needed to provide evidence of this relationship.

Lines 100-105: I think this paragraph needs to be reworked. Why is understanding how recovery time changes important and what would that mean? Why the emphasis on angling?

The hypotheses and predictions should be stated more clearly in the introduction. The authors are measuring a large number of traits, and it would be best to clearly lay out expectations to the reader.

Methods:

Lines 133-134: The authors note that there were unexpected mortalities and some individuals were exposed to multiple temperature treatments. It is unclear if this was corrected/controlled for in their analyses. From the data provided, it appears at least 9 individuals were measured under multiple temperature treatments, and if order of temperature exposure is important, then this could cause major differences in their results.

The measurements of metabolic traits raised concern. The order of measurements (maximum, post exercise, standard, and critical oxygen tension) could potentially influence the interpretation of the results. By beginning with measuring maximum metabolic rate, the authors did not acclimate the fish to the respiratory chamber or establish a relative baseline value that would be considered standard metabolic rate. Without having the standard metabolic rate baseline prior to measuring recovery, it is possible the authors are actually measuring the return to baseline values.

Additionally, the order of experiments could influence the outcomes. If possible, it would be best to know if the authors conducted preliminary tests to know whether order of operations influenced the results of the experiment.

Data analysis:

Mass was not included in the models tested, but it is known to be tightly correlated with metabolic traits.

Results:

Lines 286-299: The authors note the percent differences between current to end-of-century and current to mid-century comparisons. These percent differences are interesting, but could potentially be misleading to readers about the statistical significance of the findings. For instance, in lines 297-298, the authors state "Lastly, similar to MMR and AS, significant increases occurred in Pcrit between the current and end-of-century treatments (48%) and mid- and end- of century treatments (46%)." No p-values are provided, and the use of percentages here may create confusion as to whether or not the current and mid-century treatments are different.

Additionally, the authors seem to use mid-century to describe the 28 degree treatment, but in their introduction state that water temperatures in the Chesapeake Bay already span the 24-28 degree range. It is unclear to me how this does not also reflect current trends.

Discussion:

The discussion should be reworked to reflect changes and make grammatical corrections but is otherwise fine.

Figures:

Figure 1: Figure 1 is very complicated and is difficult to read. The information provided in this graph needs to be separated into multiple figures that could be included in the online supplement.

Figure 4: The gray lines are difficult to see. I think this could be included in the supplement as well.

Figure 5: This figure is difficult to read given the extent it relies on color for interpretation. It is not color-blind friendly. I think it could be simplified by creating multiple figures highlighting these key relationships.

Reviewer: 2

Comments to the Author(s)

GENERAL COMMENTS:

- Thank you for the opportunity to review manuscript RSOS-19704. This is a generally well-constructed draft and the paragraphs are mostly well-written with few typos which makes the review process significantly easier on a reviewer. Much appreciated.
- Though “climate change winner” is in the title – it only receives a cursory nod in the conclusion – it should be present throughout the text; introduced very early on, highlighted in the results, and interpreted in the discussion.
- Consideration of management implications is warranted as a new section. What are potential options for regulations to address anticipated impacts of post-release mortality?
- Title: smart title – should do well with search engines
- Abstract: clear and concise
- Keywords: note, for keywords, it is generally most useful to expand literature search processes by including words that are NOT in the title – so no need to state “climate change” but can add an additional relevant term, such as Chesapeake Bay
- Introduction: The content here could be much more effectively presented with extensive restructuring. I would recommend starting with importance of Chesapeake Bay and cobia, anticipated climate change (including a temperature-oxygen squeeze) within the Bay, anticipated impacts to cobia, and conclude with rationale for how to examine.
- Methods: Please clarify – were some individuals used for multiple trials? Making for non-independent samples?
- Results: See line-by-line note below about treatment labels – they are present in this section (and the abstract) but should be introduced much sooner in the text.
- Discussion: What about chronic impacts? I realize that is beyond the scope of this experiment but can you draw some interpretation based on the findings and other literature? I would also like to see more of the “climate change winner” language woven throughout this section. I also recommend breaking the “impacts of climate change and fishing” into two sections. I think doing so will allow for a more nuanced interpretation on “projected climate impacts” that will be a valuable addition to the piece.
- Conclusion: This section can “push the envelope” a bit further than just providing a summary of the results. It’s worth noting that this is the first time in the text that “climate change winner” is mentioned (though it is in the title). This conclusion could be more valuable in qualifying the

“winner” status as well as providing some discussion on management recommendations for how to address anticipated impacts.

- Table 1: need to indicate the species and why these temperature points were selected.

- Figure 1: efficient use of y-axes but a bit more spacing/separation would make them more legible. Include species name in caption?

- Figure 2, 3, 4, 5: include scientific name in caption? Also rationale for temperature selections.

- Figure 2, 3: Could consider combining y-axes as in figure 1 to create group bar charts to more simply compare the metrics across each temperature point more simply?

- Figure 5: define Ccrit in caption.

- References: not reviewed

LINE-BY-LINE COMMENTS:

40: add “mid- and end-of-century temperatures”

54-60: This paragraph could be stronger – introduce cobia, why the fishery is important, and the issue of its current status.

57: add comma after “years”; landings “have”

61-71: I would divide this into two paragraphs – one on anticipated temperature change and one on anticipated oxygen concentrations. These are the two important premises of the piece and they should establish the foundation for the study.

62: “worsen” is value-laden – consider a more objective terminology – e.g., alter? Change? Warm?

72-84: This section should build from the prior two – complex ramifications of climate change – it’s quite cursory as it currently stands and doesn’t particularly relate to cobia

77-79: how relevant to cobia? Why include?

85-99: I think this should be introduced after setting up the concern with cobia

95: comma needed after “increases”

106: transition here is somewhat abrupt between topics – consider breaking this out into its own “objectives” section?

130: some on anticipated temperatures in the Bay in the introduction is warranted to ground the decision choices here.

133: clarify – some individuals were used for multiple trials? I.e., non-independent samples? What proportion of your trials? What is the implication of that for your results and interpretation?

151: period should be inside quotation marks

168: citation for 3h period to be sufficient for microbial respiration?

224: extra space before period

251: citation for SAS?

263: spell out BIC on first mention

264: include a brief mention of what AR1 correlation structure is

289: the temperature treatments aren't introduced in the methods with the "end-of-," "mid-century," and "current monikers but they should be. Actually, even better would be to introduce them this way in the objectives.

341: evinced? Word choice?

343: if using "e.g.," "etc." is superfluous.

361: Can you extend this discussion to more chronic exposure? I realize that was not within the confines of this experiment but possibly by couching in additional literature?

363: I think this rationale is suspect, as evident by the subsequent qualification, to the point where it would be better not to include it as support for the prior statement. Unless there was some way to subset the records by when high temperatures were present but still, I think this is questionable at best.

368: but I think it goes a far way to show that the pattern is exhibited in other fishes as well.

379: define here the oxygen threshold for hypoxia in parenthetical

382: citation for climate projection?

395: reference figure?

406: take this statement further – avoidance behavior is somewhat a "catch-all" for negating projected climate impacts – elaborate on why this will be more difficult to contradict that argument.

408: This section is somewhat jumbled -- mixing interpretation of the experiment with climate projections. I think it would be clearer if it was revised into two distinct sections: 1) Post-release mortality effects and 2) projected climate impacts

412: This alternative hypothesis was not examined in this study – either reference additional research or just don't include as it wasn't a part of this work.

415-421: This is an important qualification for why cobia wouldn't actually end up a "climate change winner." I think it would be more appropriate to break this out as its own paragraph.

422-427: This jumps to implications for the fishery – I think it would be better presented later in the discussion after wrapping up the physiological impacts.

428: any other examples in the literature (cobia or other species) that found similar results?

444: Did this experiment identify temperature and oxygen thresholds? Or rather thresholds at pre-determined temperature points that correlate with expected temperatures mid- and late-century?

449: I think further qualification of the "climate change winner" status is needed – what about

prey abundance mid- and late-century? What fishing regulations could help cope with the changes?

Author's Response to Decision Letter for (RSOS-191704.R0)

See Appendix A.

RSOS-200049.R0

Review form: Reviewer 1

Is the manuscript scientifically sound in its present form?

Yes

Are the interpretations and conclusions justified by the results?

Yes

Is the language acceptable?

Yes

Do you have any ethical concerns with this paper?

No

Have you any concerns about statistical analyses in this paper?

No

Recommendation?

Accept with minor revision (please list in comments)

Comments to the Author(s)

I greatly appreciate the effort the authors put forth in revising the manuscript and addressing/incorporating reviewer comments. I find that this version of the manuscript is much improved, and my comments are mainly focused on grammatical error or awkward phrasing.

In general, the manuscript requires a review of comma usage. There are many instances where phrases are incomplete or not marked clearly by the presence of a comma, and the readability of this manuscript will improve with these edits. Similarly, there were few spelling errors that are noted in my line by line comments.

Lines 70-71: Add either "Due to a combination" or something along those lines to complete the phrase at the beginning of the sentence.

Line 72: I recommend using "we predict" rather than "we are predicted".

Line 76: Not tolerant should be intolerant.

Line 83: comma after "tolerant as their prey"

Lines 102-104: This sentence reads awkwardly. "We expect cobia to handle warm conditions" is vague, and I respect that you don't want to go into too much detail here. Maybe use withstand

instead of handle. Additionally, the phrase after the semicolon is incomplete. "it is unlikely the intense predicted to occur in coastal habitats by the end-of-century."

Line 108: comma between warm months and they are

Lines 114 and 115: possibly personal preference, but "caught and released cobia are released" reads poorly. Rephrase if possible.

Lines 121-123: How do repeated angling events on subsequent days indicate fast recovery time? I'd add a citation here.

Line 156: Comma after experimental treatments to close off the introductory phrase

Line 224: Recommend changing "After being removed" to "After removal"

Line 252: Period before SMR.

Line 308: Comma at the end of the introductory phrase

Line 408: the crit subscript for Scrit is missing the t

Lines 416-417: The sentence mentioning how impressive this tolerance is creates subjectivity. Potentially remove or rephrase to emphasize the difference between cobia and other fishes without using "impressive"

Line 448: I would remove the last sentence. Mentioning the limitations during your methods is sufficient.

Lines 480-483: I appreciate the recommendations to fisheries, but I do not feel they need be so specific given the data collected in your paper. Instead, maybe suggest broader change, such as encouraging strategies to minimize hypoxia and thermal stress.

Figure 5 still suffers from readability issues. The dark coloration creates little contrast in the upper right hand corner.

Review form: Reviewer 2

Is the manuscript scientifically sound in its present form?

Yes

Are the interpretations and conclusions justified by the results?

Yes

Is the language acceptable?

Yes

Do you have any ethical concerns with this paper?

No

Have you any concerns about statistical analyses in this paper?

No

Recommendation?

Accept with minor revision (please list in comments)

Comments to the Author(s)**GENERAL COMMENTS:**

- Thank you for the opportunity to re-review manuscript RSOS-19704. Overall, the authors have addressed my main concerns with the first draft. I've included a few minor notes below that should be considered prior to publication.
- A side note for re-reviews in the future, it is helpful for reviewers to see a track-changes version of resubmissions to more quickly locate the changes/additions in a re-review. It makes the process quicker / more streamlined.
- Management implications and a broader perspective of relevance of these findings in other contexts is still somewhat wanting.
- Introduction: Restructure is an improvement. See below for minor line-by-line comments.
- Objectives: clear as stated.
- Methods: the additional explanation on using the same individual for multiple treatments goes a long way to assuage my related concerns from the earlier draft.
- Results: Revisions are an improvement.
- Discussion: Revision to structure makes for a clearer read. However, I still would like to see a bit more of a consideration for management implications. Can the recommendations be compared with current regs? How likely are anglers to implement the suggestions?
- Conclusion: I still think this section could push the envelope a bit more. The "mortality" statement at the outset is jarring with the "climate change winner" discussion that follows – consider moving that to the end with the discussion on "compounding stressors." I would like to see one final management linkage at the end, too. And could consider the broader implications of these findings for other species in the Bay and even other recreational species more widely.
- Tables and Figures: Personally, I've always been of the impression that tables and figures should be independent of the text (e.g., a reader may just click on a figure and not read an article). So, I am still of the opinion that it is better to include the rationale for temperature, Ccrit, etc. in the figure caption.

- Figure 1: Might help address the other reviewers concerns with the y-axes by color coding them with the appropriate metabolic metric?

- References: not reviewed

LINE-BY-LINE COMMENTS:

68: spell out the mid- and end-of-century projections better here. Expand this paragraph more – it is a critical premise for the study.

76: complex paragraph – consider simplifying and maybe breaking out into two.

79: I think just the reference is suitable – it's kind of confusing to see smelt called out here – in a quick read, I was thinking how are smelt related to cobia?!?

95: comma needed after “ changes”

116: high oxygen and optima temperatures do NOT promote recovery?!?

394: this sentence seems weak as written – would be stronger to state that “Without further studies on the physiology of cobia, such as quantifying the effect of temperature on blood oxygen binding, gill gas exchange rates, and heart function, the patterns in AS we observed cannot be experimentally supported” or something like that...

397: extra space before period.

434: “whatever conditions”? Reconsider word choice.

437: Had to read this sentence a few times – maybe revise to simplify/clarify.

461: Swap this paragraph down to the next section on impacts of fishing

480: break this out into its own paragraph and expand here – are these feasible recommendations? Provide context with the current regs. Are cobia anglers familiar with these post-capture methods? When is the current fishing season and how much would you recommend it change?

488: elaborate on why exhaustive exercise can result in mortality. There is a disjunct between these two sentences – stating that there can be mortality and then calling cobia a winner? I think the order of these statements needs revisiting.

497: any management relevant conclusions to add? What are the broader implications of these findings for other Bay species? Other recreational species?

Decision letter (RSOS-200049.R0)

18-Feb-2020

Dear Mr Crear

On behalf of the Editor, I am pleased to inform you that your Manuscript RSOS-200049 entitled "A climate change winner: Measuring the effects of warming and hypoxia on the metabolism of an important recreational fish species" has been accepted for publication in Royal Society Open Science subject to minor revision in accordance with the referee suggestions. Please find the referees' comments at the end of this email.

The reviewers and Subject Editor have recommended publication, but also suggest some minor revisions to your manuscript. Therefore, I invite you to respond to the comments and revise your manuscript.

- Ethics statement

- Data accessibility

<http://datadryad.org/submit?journalID=RSOS&manu=RSOS-200049>

- Competing interests

- Authors' contributions

- Acknowledgements

- Funding statement

Because the schedule for publication is very tight, it is a condition of publication that you submit the revised version of your manuscript before 27-Feb-2020. Please note that the revision deadline will expire at 00.00am on this date. If you do not think you will be able to meet this date please let me know immediately.

To revise your manuscript, log into <https://mc.manuscriptcentral.com/rsos> and enter your Author Centre, where you will find your manuscript title listed under "Manuscripts with Decisions". Under "Actions," click on "Create a Revision." You will be unable to make your

revisions on the originally submitted version of the manuscript. Instead, revise your manuscript and upload a new version through your Author Centre.

on behalf of Dr Michael Tobler (Associate Editor) and Kevin Padian (Subject Editor)
openscience@royalsociety.org

Associate Editor Comments to Author (Dr Michael Tobler):

We have received the evaluations from both of the previous reviewers. Both commend the authors for their revisions, but also highlight a number of additional points. I think these suggestions can be addressed with some minor revisions after which the manuscript is acceptable for publication.

Reviewer comments to Author:

Reviewer: 2

Comments to the Author(s)

GENERAL COMMENTS:

- Thank you for the opportunity to re-review manuscript RSOS-19704. Overall, the authors have addressed my main concerns with the first draft. I've included a few minor notes below that should be considered prior to publication.

- A side note for re-reviews in the future, it is helpful for reviewers to see a track-changes version of resubmissions to more quickly locate the changes/additions in a re-review. It makes the process quicker / more streamlined.

- Management implications and a broader perspective of relevance of these findings in other contexts is still somewhat wanting.

- Introduction: Restructure is an improvement. See below for minor line-by-line comments.

- Objectives: clear as stated.

- Methods: the additional explanation on using the same individual for multiple treatments goes a long way to assuage my related concerns from the earlier draft.

- Results: Revisions are an improvement.

- Discussion: Revision to structure makes for a clearer read. However, I still would like to see a bit more of a consideration for management implications. Can the recommendations be compared with current regs? How likely are anglers to implement the suggestions?

- Conclusion: I still think this section could push the envelope a bit more. The "mortality" statement at the outset is jarring with the "climate change winner" discussion that follows – consider moving that to the end with the discussion on "compounding stressors." I would like to see one final management linkage at the end, too. And could consider the broader implications of these findings for other species in the Bay and even other recreational species more widely.

- Tables and Figures: Personally, I've always been of the impression that tables and figures should be independent of the text (e.g., a reader may just click on a figure and not read an article). So, I am still of the opinion that it is better to include the rationale for temperature, Ccrit, etc. in the figure caption.

- Figure 1: Might help address the other reviewers concerns with the y-axes by color coding them with the appropriate metabolic metric?

- References: not reviewed

LINE-BY-LINE COMMENTS:

68: spell out the mid- and end-of-century projections better here. Expand this paragraph more – it is a critical premise for the study.

76: complex paragraph – consider simplifying and maybe breaking out into two.

79: I think just the reference is suitable – it's kind of confusing to see smelt called out here – in a quick read, I was thinking how are smelt related to cobia???

95: comma needed after "changes"

116: high oxygen and optima temperatures do NOT promote recovery?!?

394: this sentence seems weak as written – would be stronger to state that "Without further studies on the physiology of cobia, such as quantifying the effect of temperature on blood oxygen binding, gill gas exchange rates, and heart function, the patterns in AS we observed cannot be experimentally supported" or something like that...

397: extra space before period.

434: "whatever conditions"? Reconsider word choice.

437: Had to read this sentence a few times – maybe revise to simplify/clarify.

461: Swap this paragraph down to the next section on impacts of fishing

480: break this out into its own paragraph and expand here – are these feasible recommendations? Provide context with the current regs. Are cobia anglers familiar with these post-capture methods? When is the current fishing season and how much would you recommend it change?

488: elaborate on why exhaustive exercise can result in mortality. There is a disjunct between these two sentences – stating that there can be mortality and then calling cobia a winner? I think the order of these statements needs revisiting.

497: any management relevant conclusions to add? What are the broader implications of these findings for other Bay species? Other recreational species?

Reviewer: 1

Comments to the Author(s)

I greatly appreciate the effort the authors put forth in revising the manuscript and addressing/incorporating reviewer comments. I find that this version of the manuscript is much improved, and my comments are mainly focused on grammatical error or awkward phrasing.

In general, the manuscript requires a review of comma usage. There are many instances where phrases are incomplete or not marked clearly by the presence of a comma, and the readability of this manuscript will improve with these edits. Similarly, there were few spelling errors that are noted in my line by line comments.

Lines 70-71: Add either "Due to a combination" or something along those lines to complete the phrase at the beginning of the sentence.

Line 72: I recommend using "we predict" rather than "we are predicted".

Line 76: Not tolerant should be intolerant.

Line 83: comma after "tolerant as their prey"

Lines 102-104: This sentence reads awkwardly. "We expect cobia to handle warm conditions" is vague, and I respect that you don't want to go into too much detail here. Maybe use withstand instead of handle. Additionally, the phrase after the semicolon is incomplete. "it is unlikely the intense predicted to occur in coastal habitats by the end-of-century."

Line 108: comma between warm months and they are

Lines 114 and 115: possibly personal preference, but "caught and released cobia are released" reads poorly. Rephrase if possible.

Lines 121-123: How do repeated angling events on subsequent days indicate fast recovery time? I'd add a citation here.

Line 156: Comma after experimental treatments to close off the introductory phrase

Line 224: Recommend changing "After being removed" to "After removal"

Line 252: Period before SMR.

Line 308: Comma at the end of the introductory phrase

Line 408: the crit subscript for Scrit is missing the t

Lines 416-417: The sentence mentioning how impressive this tolerance is creates subjectivity. Potentially remove or rephrase to emphasize the difference between cobia and other fishes without using "impressive"

Line 448: I would remove the last sentence. Mentioning the limitations during your methods is sufficient.

Lines 480-483: I appreciate the recommendations to fisheries, but I do not feel they need be so specific given the data collected in your paper. Instead, maybe suggest broader change, such as encouraging strategies to minimize hypoxia and thermal stress.

Figure 5 still suffers from readability issues. The dark coloration creates little contrast in the upper right hand corner.

Author's Response to Decision Letter for (RSOS-200049.R0)

See Appendix B.

Decision letter (RSOS-200049.R1)

24-Feb-2020

Dear Mr Crear,

It is a pleasure to accept your manuscript entitled "In the face of climate change and exhaustive exercise: The physiological response of an important recreational fish species" in its current form for publication in Royal Society Open Science. The comments of the reviewer(s) who reviewed your manuscript are included at the foot of this letter.

Please can you supply the Editorial Office with the 'for review' version of your Dryad data

deposition URL? This will be used in the published version of the manuscript, rather than the current 'for review' version.

on behalf of Dr Michael Tobler (Associate Editor) and Kevin Padian (Subject Editor)
openscience@royalsociety.org

Appendix A

Dear Dr. Michael Tobler & Kevin Padian

Enclosed you will find our revised manuscript entitled "A climate change winner: Measuring the effects of warming and hypoxia on the metabolism of an important recreational fish species." We appreciate the reviewers' feedback and believe we have addressed all their concerns and comments. Below we address specific comments of each reviewer. Corrections that have been made and incorporated into the revised manuscript are shown in red below. We believe this paper will make a significant contribution to the field of marine research. We look forward to hearing from you.

Editor Comments

- Figure 1: Avoid graphs that plot multiple dependent variables and have multiple y-axes. Instead, create multi-panel figure where each dependent variable is plotted on top of the other. That still allows the reader to detect trends across variables, but it's much easier to figure out what data points belong to what axis.

Addressed this further down in response to the reviewer.

-Figures 2 and 3: Instead of plotting mean and CI, consider a form of data visualization that shows the data (or at least data distribution). See here for more information: <https://journals.plos.org/plosbiology/article?id=10.1371/journal.pbio.1002128>

We understand the editors concerns, but because we are discussing model results throughout the results section we believe it is most appropriate to display the model results in the figures which is just the model estimate (mean) and variation around the mean (CI) for each temperature treatment. To display the data distribution we had included box plots in the supplementary material. To clarify this we added this line at the beginning of the results section:

"Referenced model response variables (e.g. SMR, AS, metabolic rate) below represent model estimates, while raw values are indicated as such."

-Figure 5: Please consider alternative color scales to make the figure easier to interpret: See here for more information: <https://journals.plos.org/plosone/article?id=10.1371/journal.pone.0199239>

We used the link below to help us adjust the colors of this figure.

Reviewer 1:

Comments to the Author(s)

This study sought to address the vulnerability of an economically important fish species to global climate change scenarios through a physiological lens. Without a doubt, understanding species-specific responses to climate change is important, especially in threatened or economically beneficial species, but I do not think the authors were able to successfully support their claims in this submission. My comments are as follows:

Introduction:

The introduction struggles to find a balance between discussing the consequences of stress from recreational (angling) and climate change (temperature and hypoxia) sources.

Content was added and the structure was rearranged based on the this comment and the second reviewer comments. See the second reviewer's comments for a more detailed response.

Lines 98-99: A citation is needed to provide evidence of this relationship.

Citation added.

Lines 100-105: I think this paragraph needs to be reworked. Why is understanding how recovery time changes important and what would that mean? Why the emphasis on angling?

Made adjustments to this paragraph and added the following to emphasize why understanding changes in recovery time is important.

“After being stressed through capture, handling, and air exposure, caught and released cobia may be released into environmentally stressful conditions. Being released into conditions that do not promote recovery (e.g. high oxygen and optimal temperatures) could prolong recovery and reduce their chances of survival post release.... and help inform managers how well cobia may recover from capture post release under increasing temperatures expected to occur in coastal habitats.”

The hypotheses and predictions should be stated more clearly in the introduction. The authors are measuring a large number of traits, and it would be best to clearly lay out expectations to the reader.

Because we are measuring a handful of traits and based on the second reviewer's comments we decided to make the objectives paragraph a separate section. We felt that leaving this information as objectives would suffice instead of having a separate hypothesis for each trait. We did however add some predictions throughout the introduction regarding each trait.

“Being exposed to subtropical waters, we expect cobia to handle warm conditions, however, it is unlikely the intense predicted to occur in coastal habitats by the end-of-century.”

“Due to cobia's exposure to low oxygen when they use bays and estuaries in the warm months they are expected to be relatively hypoxia tolerant.”

“We predict that like other species, cobia ventilation volume will track with their metabolic rate.”

“As conditions become more stressful (i.e. warmer) we expect recovery time to increase. However, unpublished catch and released data show the same cobia being caught on subsequent days, which suggest that they will have a fast recovery time.”

Methods:

Lines 133-134: The authors note that there were unexpected mortalities and some individuals were exposed to multiple temperature treatments. It is unclear if this was corrected/controlled for in their analyses. From the data provided, it appears at least 9 individuals were measured under

multiple temperature treatments, and if order of temperature exposure is important, then this could cause major differences in their results.

Expanded a lot on this section. Due to a facility system failure in our seawater lab we had a large mortality event occur in our holding tank in between treatments. We therefore had to get a second set of cobia the following summer to conduct the remaining treatments. We addressed the concerns about using the same individual in the Data Analysis section. Below is what we added:

“Because of mortalities suffered in the holding tank between experimental treatments no individuals were tested at all three temperature treatments and nine were tested at two temperatures. Of the 13 cobia tested at 24°C, one was also tested at 28°C. Eight individuals were tested at both 28° and 32°C. The issue of using the same individual for multiple treatments is addressed in the Data Analysis section below.”

A sentence in the Data Analysis section was added indicating that we initially included individual as a random effect in the model but it made no difference to the model so we decided to be parsimonious and leave it out in our final model. The sentence we added below:

“A random effect of individual was initially included in the model to account for using the same individual for multiple treatments, but the model did not change; therefore to ensure parsimony the random effect was not used in the final model.”

The measurements of metabolic traits raised concern. The order of measurements (maximum, post exercise, standard, and critical oxygen tension) could potentially influence the interpretation of the results. By beginning with measuring maximum metabolic rate, the authors did not acclimate the fish to the respiratory chamber or establish a relative baseline value that would be considered standard metabolic rate. Without having the standard metabolic rate baseline prior to measuring recovery, it is possible the authors are actually measuring the return to baseline values.

Measuring MMR, SMR, then Scrit is a common order of measurements in the literature. Please see Slesinger et al. 2019, Crear et al. 2019, Lapointe et al. 2014, Marras et al. 2013, etc. In addition, Marras et al. 2013 and other studies used similar approach to measure recovery time. Lastly, because cobia are extremely powerful and strong and our cobia ranged between 81-108 cm TL and 3-10 kg we wanted to reduce handling time to avoid injury to the animals. We actually realized that we had not provided the size of our fish during the entire manuscript so we added that information in the Methods. So unlike many respirometry studies where smaller fish were used and could more easily be manipulated, the large size of our fish limited our ability to put the animal in the respirometer, get it's SMR first then take it out, chase it in our separate chase tank, then put it back into our respirometer to get MMR and recovery time. We also made sure that the metabolic rate completely leveled off and stabilized before we considered it to be SMR. The leveling off is indicated in Figure 1. To make the reader aware that the authors are aware of this limitation was added this sentence after the Maximum and Standard Metabolic Rate section in the methods:

“This method of determining SMR was selected because the large size of our fish inhibited us from manipulating them too much and thus measuring a SMR baseline prior to the fish undergoing the chase protocol.”

We also added these sentence at the end of the recovery section in the Discussion:

“Lastly, it is important to note that by measuring SMR (baseline) after the fish appeared to be recovered from the chase protocol instead of getting a baseline first, we may have overestimated

SMR and underestimated recovery time. We tried to overcome this by waiting until the fish's metabolic rate stabilized.”

Additionally, the order of experiments could influence the outcomes. If possible, it would be best to know if the authors conducted preliminary tests to know whether order of operations influenced the results of the experiment.

Please see our above response that the size of our animals made it very difficult to manipulate them so we were unable to conduct preliminary tests.

Data analysis:

Mass was not included in the models tested, but it is known to be tightly correlated with metabolic traits.

Mass is included in metabolic rate calculation (see metabolic rate equation in Methods) so there is no need to include this in the model.

Results:

Lines 286-299: The authors note the percent differences between current to end-of-century and current to mid-century comparisons. These percent differences are interesting, but could potentially be misleading to readers about the statistical significance of the findings. For instance, in lines 297-298, the authors state "Lastly, similar to MMR and AS, significant increases occurred in Pcrit between the current and end-of-century treatments (48%) and mid- and end- of century treatments (46%)." No p-values are provided, and the use of percentages here may create confusion as to whether or not the current and mid-century treatments are different.

Throughout the text there is reference to the Table 1 which provides the p-values generated from the a priori contrast statements of least square means.

Additionally, the authors seem to use mid-century to describe the 28 degree treatment, but in their introduction state that water temperatures in the Chesapeake Bay already span the 24-28 degree range. It is unclear to me how this does not also reflect current trends.

In the second paragraph of the methods it states “Experimental temperatures were selected to represent commonly occurring summertime temperatures in Chesapeake Bay (24 and 28°C, with the latter expected to occur more often by mid-century), and a temperature (32°C) that rarely occurs today but that is expected to occur more often by end-of-century.” So although 28° occurs currently due to variation in temperature common in coastal habitats, it is on the higher end of temperatures that we see in Chesapeake Bay and we expect it to be more common in the future. In addition, because we needed to select single temperatures for treatments we had to decide a representative temperature for each time period despite the fact that we will see some overlap in these temperatures in each time period.

Discussion:

The discussion should be reworked to reflect changes and make grammatical corrections but is otherwise fine.

Please see changes regarding the discussion in response to the second reviewer's comments.

Figures:

Figure 1: Figure 1 is very complicated and is difficult to read. The information provided in this graph needs to be separated into multiple figures that could be included in the online supplement.

We felt that the current figure needs to stay in its current form because when separated out into separate figures we is extremely difficult to discern patterns overtime or the scope of each dependent variable because the yaxis is shrunk too much. Please see the graph below:

The authors understand the figure is complicated but we feel it is the best way to compare trends throughout the trial. We are happy to adjust colors to optimize visual needs. We need thicken the lines to make it easier to see. Lastly, the other reviewer liked our use of yaxes in figure 1.

Figure 4: The gray lines are difficult to see. I think this could be included in the supplement as well.

Grey lines were darkened and thickened to make the figure easier to see. The authors believe this figure should remain in the main text because it helps the reader understand the relationship between ventilation and metabolic rate in normoxic conditions.

Figure 5: This figure is difficult to read given the extent it relies on color for interpretation. It is not color-blind friendly. I think it could be simplified by creating multiple figures highlighting these key relationships.

We felt this this figure showed the relationship between three variables and that we would lose information by separating the data out into multiple figures. We did however change the colors so that it would be color-blind friendly based on the article provided by the editor.

Reviewer 2

Comments to the Author(s)

GENERAL COMMENTS:

Addressed these general comments through

- Thank you for the opportunity to review manuscript RSOS-19704. This is a generally well-constructed draft and the paragraphs are mostly well-written with few typos which makes the review process significantly easier on a reviewer. Much appreciated.
- Though “climate change winner” is in the title – it only receives a cursory nod in the conclusion – it should be present throughout the text; introduced very early on, highlighted in the results, and interpreted in the discussion.

See responses in line by line response.

- Consideration of management implications is warranted as a new section. What are potential options for regulations to address anticipated impacts of post-release mortality?

See responses in line by line response.

- Title: smart title – should do well with search engines

- Abstract: clear and concise

- Keywords: note, for keywords, it is generally most useful to expand literature search processes by including words that are NOT in the title – so no need to state “climate change” but can add an additional relevant term, such as Chesapeake Bay

Good idea, changed climate change to Chesapeake Bay in the keywords.

- Introduction: The content here could be much more effectively presented with extensive restructuring. I would recommend starting with importance of Chesapeake Bay and cobia, anticipated climate change (including a temperature-oxygen squeeze) within the Bay, anticipated impacts to cobia, and conclude with rationale for how to examine.

See responses in line by line response.

- Methods: Please clarify – were some individuals used for multiple trials? Making for non-independent samples?

See responses in line by line response.

- Results: See line-by-line note below about treatment labels – they are present in this section (and the abstract) but should be introduced much sooner in the text.

Add to the abstract. Also see responses in line by line response.

- Discussion: What about chronic impacts? I realize that is beyond the scope of this experiment but can you draw some interpretation based on the findings and other literature? I would also like to see more of the “climate change winner” language woven throughout this section. I also

recommend breaking the “impacts of climate change and fishing” into two sections. I think doing so will allow for a more nuanced interpretation on “projected climate impacts” that will be a valuable addition to the piece.

See responses in line by line response.

- Conclusion: This section can “push the envelope” a bit further than just providing a summary of the results. It’s worth noting that this is the first time in the text that “climate change winner” is mentioned (though it is in the title). This conclusion could be more valuable in qualifying the “winner” status as well as providing some discussion on management recommendations for how to address anticipated impacts.

See responses in line by line response.

- Table 1: need to indicate the species and why these temperature points were selected.

Added the species name to the table caption. The selected temperature information is detailed in the intro and methods. We felt all of that information did not need to be repeated in the table caption.

- Figure 1: efficient use of y-axes but a bit more spacing/separation would make them more legible. Include species name in caption?

Added species name in the figure caption. Created more space between yaxes.

- Figure 2, 3, 4, 5: include scientific name in caption? Also rationale for temperature selections.

Added scientific name after cobia in each figure caption. As mentioned above the temperature rationale is in the intro and methods.

- Figure 2, 3: Could consider combining y-axes as in figure 1 to create group bar charts to more simply compare the metrics across each temperature point more simply?

Although this would simplify the graphs, we think that by combining the three graphs would affect the reader’s ability to discern differences within each metric among the different temperatures.

- Figure 5: define Ccrit in caption.

This is clearly defined in the methods section. We felt all of that information did not need to be repeated in the figure caption.

- References: not reviewed

LINE-BY-LINE COMMENTS:

40: add “mid- and end-of-century temperatures”

Correction made.

54-60: This paragraph could be stronger – introduce cobia, why the fishery is important, and the issue of its current status.

Added more information and sources about the importance of the cobia fishery and reworded the paragraph:

“With over 225,000 trips per year targeting cobia in Virginia alone and anglers valuing cobia fishing between \$488-\$685 per trip, this recreational fishery is important for U.S. coastal states [2]. Cobia are particularly vulnerable and heavily fished when they migrate into high salinity bays and estuaries, such as Chesapeake Bay to spawn between May-September [3-5]. In recent years, landings of cobia along the U.S. East Coast has exceeded the annual catch limit by over 200% [6]. This has led to the early or complete closure of the fishery in federal waters [7, 8].”

57: add comma after “years”; landings “have”

Correction made.

61-71: I would divide this into two paragraphs – one on anticipated temperature change and one on anticipated oxygen concentrations. These are the two important premises of the piece and they should establish the foundation for the study.

Good idea. Separated this out into two paragraphs.

62: “worsen” is value-laden – consider a more objective terminology – e.g., alter? Change? Warm?

Used “warm”

72-84: This section should build from the prior two – complex ramifications of climate change – it’s quite cursory as it currently stands and doesn’t particularly relate to cobia

Linked cobia to this paragraph and rearranged the paragraph as by adding:

“In a warming and increasingly hypoxic Chesapeake Bay, fish species not tolerant of these conditions may have to shift their habitat selection. These shifts can result in a spatial mismatch between fish and their prey [14]. This becomes a particular problem for fish species (e.g., smelt, *Osmerus eperlanus*; Horppila et al., 2000) that prey on benthic invertebrates, which can often tolerate lower oxygen levels than many fishes [14, 16]. Although cobia use the entire water column, two of their most common prey items, blue crabs (*Callinectes sapidus*) and Atlantic croaker (*Micropogonias undulates*) use benthic habitats [17]; therefore, if cobia are not as hypoxia tolerant as their prey their diet or distribution inside these coastal habitats may shift. Another phenomenon occurring as a result of warming and increasing hypoxia is called the “temperature-oxygen squeeze” [18], where hypoxic bottom waters force fish to seek out more oxygenated, but warmer, surface waters. As a result, fish can be exposed to physiologically stressful temperatures or dissolved oxygen levels [14, 18, 19]. Further, the temperature-oxygen squeeze has forced species like striped bass (*Morone saxatilis*) in Chesapeake Bay to use warmer surface waters, which has increased their susceptibility to diseases, like *Mycobacterium* [19]. If cobia are negatively impacted by changing conditions we may observe shifts in species distribution, which can lead to changes in migration [20, 21], reproductive patterns [22], and abundance levels [22, 23], ultimately influencing the population level success.”

77-79: how relevant to cobia? Why include?

See above.

85-99: I think this should be introduced after setting up the concern with cobia

We believe that the physiological information should be after the climate change information because we directly link physiology and metrics measured to changes going on as a result of climate change.

95: comma needed after “increases”

Correction made.

106: transition here is somewhat abrupt between topics – consider breaking this out into its own “objectives” section?

Separated this paragraph out into its own Objective section.

130: some on anticipated temperatures in the Bay in the introduction is warranted to ground the decision choices here.

This information is above in the climate change paragraph in Chesapeake Bay.

“The most common temperatures throughout the entire water column within Chesapeake Bay during the summer spans 24-28°C. With climate change, the water temperatures in Chesapeake Bay are expected to increase between 1.5 and 2°C by the mid-century and by 5°C by the end-of-century relative to the mid 1990s.”

133: clarify – some individuals were used for multiple trials? I.e., non-independent samples? What proportion of your trials? What is the implication of that for your results and interpretation?

Expanded a lot on this section. Due to a facility system failure in our seawater lab we had a large mortality event occur in our holding tank in between treatments. We therefore had to get a second set of cobia the following summer to conduct the remaining treatments. We addressed the concerns about using the same individual in the Data Analysis section. Below is what we added:

“Because of mortalities suffered in the holding tank between experimental treatments no individuals were tested at all three temperature treatments and nine were tested at two temperatures. Of the 13 cobia tested at 24°C, one was also tested at 28°C. Eight individuals were tested at both 28° and 32°C. The issue of using the same individual for multiple treatments is addressed in the Data Analysis section below.”

A sentence in the Data Analysis section was added indicating that we initially included individual as a random effect in the model but it made no difference to the model so we decided to be parsimonious and leave it out in our final model. The sentence we added below:

“A random effect of individual was initially included in the model to account for using the same individual for multiple treatments, but the model did not change; therefore to ensure parsimony the random effect was not used in the final model.”

151: period should be inside quotation marks

Correction made.

168: citation for 3h period to be sufficient for microbial respiration?

Citation added.

224: extra space before period

Good catch. Removed the period.

251: citation for SAS?

Added a citation for paper that did similar analyses in SAS.

263: spell out BIC on first mention

Correction made.

264: include a brief mention of what AR1 correlation structure is

This type of description would unfortunately not be brief so we feel that this should not be explained in this paper. We did write out what AR1 stood for, which is autoregressive of order 1.

289: the temperature treatments aren't introduced in the methods with the "end-of-," "mid-century," and "current monikers but they should be. Actually, even better would be to introduce them this way in the objectives.

Added all three monikers in the objectives and at the beginning of the methods.

341: evinced? Word choice?

Instead of evinced used "supported."

343: if using "e.g.," "etc." is superfluous.

Correction made.

361: Can you extend this discussion to more chronic exposure? I realize that was not within the confines of this experiment but possibly by couching in additional literature?

Added information in the climate change section address this a little more:

"Their adaptations to the variable environmental conditions in coastal habitats appear to allow cobia to withstand high temperatures and low ambient oxygen levels. The lack of mortality or change in behavior while in the holding tank for over a month at the end-of-century treatment suggests that chronic exposure to high temperatures may not be detrimental to this species."

363: I think this rationale is suspect, as evident by the subsequent qualification, to the point where it would be better not to include it as support for the prior statement. Unless there was some way to subset the records by when high temperatures were present but still, I think this is questionable at best.

Good point. We removed that section.

368: but I think it goes a far way to show that the pattern is exhibited in other fishes as well.

Agreed, meaning that there is a lot more going on with AS and may be determined by quantifying the effect of temperature on blood oxygen binding, gill gas exchange rates, and heart function

379: define here the oxygen threshold for hypoxia in parenthetical

The value is rather subjective and species and field dependent, but in the context of coastal hypoxia for species we selected <2mg/L.

382: citation for climate projection?

Added a citation.

395: reference figure?

We ended up removing this sentence.

406: take this statement further – avoidance behavior is somewhat a “catch-all” for negating projected climate impacts – elaborate on why this will be more difficult to contradict that argument.

Avoidance has been and will continue to be a major animal response to climate change, but this is a good point for cobia because although cobia could just move, they rely on Chesapeake Bay for spawning. Adjusted the end of this paragraph:

“Although cobia are likely to avoid areas with this combination of stressful conditions in Chesapeake Bay (i.e. elevated temperatures and reduced oxygen levels), avoidance may become more difficult as conditions worsen within Chesapeake Bay during the summer months due to the effects of climate change, particularly by end-of-century. Due to the importance of Chesapeake Bay to cobia spawning, fish may have to sustain whatever conditions to ensure spawning occurs.”

408: This section is somewhat jumbled -- mixing interpretation of the experiment with climate projections. I think it would be clearer if it was revised into two distinct sections: 1) Post-release mortality effects and 2) projected climate impacts

We agree with this comment and added a separate Recovery Post-Exercise section to discuss physiology what may be going on before the climate change section. Then we also added an Impacts of Fishing section to discuss implications of recovery on the fishery.

412: This alternative hypothesis was not examined in this study – either reference additional research or just don’t include as it wasn’t a part of this work.

We think that this hypothesis could totally occur based on what we found. Although cobia are tolerant of current and future conditions, if they can adjust their habitat use spatially or temporally and maintain the necessary requirements for spawning/feeding and be in conditions they prefer (not just tolerate) then shifts could certainly occur. This type of speculation is directly tied to climate change impacts on cobia and is thus related to the results of this study. Based on this we feel that we should keep this sentence.

415-421: This is an important qualification for why cobia wouldn’t actually end up a “climate change winner.” I think it would be more appropriate to break this out as its own paragraph.

Created a second paragraph for this.

422-427: This jumps to implications for the fishery – I think it would be better presented later in the discussion after wrapping up the physiological impacts.

As mentioned above, we created a separate fishing impacts section to discuss this after the physiology explanations.

428: any other examples in the literature (cobia or other species) that found similar results?

No unfortunately not.

444: Did this experiment identify temperature and oxygen thresholds? Or rather thresholds at pre-determined temperature points that correlate with expected temperatures mid- and late-century?

Reworded and said:

“We identified cobia’s response to predicted conditions expected to occur in Chesapeake Bay.”

449: I think further qualification of the “climate change winner” status is needed – what about prey abundance mid- and late-century? What fishing regulations could help cope with the changes?

We address that cobia may be a climate change winner in the climate change section and in the conclusion. Added a sentence about prey, see below:

“Further, being a generalist feeder and selecting prey items that also are hypoxia tolerant will benefit cobia in the future when their prey are still present despite worsening conditions.”

We also added information about potential fishing regulations to the fishing impacts section.

Appendix B

Dear Dr. Michael Tobler & Kevin Padian

Our revised manuscript entitled "In the face of climate change and exhaustive exercise: The physiological response of an important recreational fish species" has been uploaded. We appreciate the additional reviewers' feedback and believe their comments improved the manuscript as a whole. We have addressed each reviewer's concerns and comments below. Corrections that have been made and incorporated into the revised manuscript are shown in red. We believe this revised paper will make a significant contribution at the intersection of climate change and fish physiology research. We look forward to hearing from you.

Based on the comments of one of the reviewers during the first round of reviews and again on this second round, the authors felt that we should adjust the title a little bit and remove the climate change winner part. As the reviewer pointed out, we think it is a stretch to generalize cobia as a climate change winner because at the warmest temperatures, when some cobia were exercised they suffered a mortality. Although cobia will likely be able to tolerate conditions better than many other species, we think it is a stretch to claim that cobia as a whole will be a climate change winner. We would argue that by mid-century cobia would certainly be considered a winner but when we stretch to end-of-century, the story slightly changes. Therefore, we generalized our title a little to encompass both stressors, climate change and exhaustive exercise, and switched "metabolism" with "physiological response" because in addition to metabolism we also measure ventilation and recovery time. Anywhere we referred to cobia as a climate change winner, we specified for mid-century or near future.

Please ensure the initial submission of the raw data in Dryad follows this manuscript, as that material is

Reviewer comments to Author:

Reviewer: 2

Comments to the Author(s)

GENERAL COMMENTS:

- Thank you for the opportunity to re-review manuscript RSOS-19704. Overall, the authors have addressed my main concerns with the first draft. I've included a few minor notes below that should be considered prior to publication.

- A side note for re-reviews in the future, it is helpful for reviewers to see a track-changes version of resubmissions to more quickly locate the changes/additions in a re-review. It makes the process quicker / more streamlined.

- Management implications and a broader perspective of relevance of these findings in other contexts is still somewhat wanting.

Addressed this in the specific comments below.

- Introduction: Restructure is an improvement. See below for minor line-by-line comments.

- Objectives: clear as stated.
- Methods: the additional explanation on using the same individual for multiple treatments goes a long way to assuage my related concerns from the earlier draft.
- Results: Revisions are an improvement.
- Discussion: Revision to structure makes for a clearer read. However, I still would like to see a bit more of a consideration for management implications. Can the recommendations be compared with current regs? How likely are anglers to implement the suggestions?
Addressed this in the specific comments below.
- Conclusion: I still think this section could push the envelope a bit more. The “mortality” statement at the outset is jarring with the “climate change winner” discussion that follows – consider moving that to the end with the discussion on “compounding stressors.” I would like to see one final management linkage at the end, too. And could consider the broader implications of these findings for other species in the Bay and even other recreational species more widely.
Addressed this in the specific comments below.
- Tables and Figures: Personally, I’ve always been of the impression that tables and figures should be independent of the text (e.g., a reader may just click on a figure and not read an article). So, I am still of the opinion that it is better to include the rationale for temperature, Ccrit, etc. in the figure caption.
Since it appears this comment is based on personal preference, the authors believe the current information in the captions should suffice. We also think this avoids repetitive text.
- Figure 1: Might help address the other reviewers concerns with the y-axes by color coding them with the appropriate metabolic metric?
Good idea. We color coded the axes to match the line colors.
- References: not reviewed

LINE-BY-LINE COMMENTS:

68: spell out the mid- and end-of-century projections better here. Expand this paragraph more – it is a critical premise for the study.

Added a sentence to the end of the paragraph to justify the temperature treatment selections. “Therefore, as temperatures warm we expect 28°C to occur even more frequently by mid-century and 32°C water temperatures to be very common by the end-of-century.”

76: complex paragraph – consider simplifying and maybe breaking out into two.

We do think all the material in the paragraph is necessary for the reading to fully understand the impacts of climate change on species in Chesapeake Bay. But we did split the paragraph into two paragraphs where the discussion about the temperature-oxygen squeeze is introduced.

79: I think just the reference is suitable – it's kind of confusing to see smelt called out here – in a quick read, I was thinking how are smelt related to cobia?!?

Good point. Removed the text that said smelt.

95: comma needed after “changes”

Comma added.

116: high oxygen and optima temperatures do NOT promote recovery?!?

Good catch. Changed it to “low oxygen and non-optimal temperatures”

394: this sentence seems weak as written – would be stronger to state that “Without further studies on the physiology of cobia, such as quantifying the effect of temperature on blood oxygen binding, gill gas exchange rates, and heart function, the patterns in AS we observed cannot be experimentally supported” or something like that...

We think that using the terms “difficult to explain” suffices since those are all reasons that could help explain the patterns we saw that are currently difficult to explain based on what we measured. We think by putting “cannot be experimentally supported” is rather confusing because by conducting those other experiments/metrics it could support out findings. We are also unsure what the reviewer means by “experimentally support.”

397: extra space before period.

Correction made.

434: “whatever conditions”? Reconsider word choice.

Switched it to “endure less suitable conditions.”

437: Had to read this sentence a few times – maybe revise to simplify/clarify.

Clarified and put into two sentence. “Typically as AS increases, recovery time following exhaustive exercise is shorter [44], but we actual found the opposite for the warmest temperatures. In addition, recovery time was shorter at high temperatures when AS was higher (for 32°C).”

461: Swap this paragraph down to the next section on impacts of fishing

Added this paragraph to the beginning of the Fishing Impacts section.

480: break this out into its own paragraph and expand here – are these feasible recommendations? Provide context with the current regs. Are cobia anglers familiar with these post-capture methods? When is the current fishing season and how much would you recommend it change?

Broke this section out into its own paragraph and added a couple sentences to this so it now looks like this: “To improve future post release survival as conditions worsen, fisheries management could encourage leaving the fish in the water post capture prior to release. Currently all cobia caught are typically brought into the boat, dehooked, and are either kept or released. Regulations have been put into place to leave some important recreational fish species (e.g. white marlin; [65]) in the water. Further, by limiting the fishing season to months that are less warm (e.g., May, June and September), we would avoid releasing stressed fish into

suboptimal conditions. Although these recommendations are premature for today's conditions, we do believe they may reduce mortality in the coming decades."

488: elaborate on why exhaustive exercise can result in mortality. There is a disjunct between these two sentences – stating that there can be mortality and then calling cobia a winner? I think the order of these statements needs revisiting.

Ended up removing the part of the sentence that mentions mortality so that there is a smoother transition to the next sentence. I also added "in the near future (30+ years)" after "climate change winner" to indicate that they will be fine for mid-century. Discussion of excess fishing stress is mentioned further down in the paragraph. Also we removed winner from abstract and indicated more specifically that they can withstand near future conditions. We added similar language to other text when climate change winner was mentioned.

497: any management relevant conclusions to add? What are the broader implications of these findings for other Bay species? Other recreational species?

Added a sentence at the very end, "This type of information can be useful to future managers who will have to consider the interactions among fish species, the changing environment, and various fishing practices."

Reviewer: 1

Comments to the Author(s)

I greatly appreciate the effort the authors put forth in revising the manuscript and addressing/incorporating reviewer comments. I find that this version of the manuscript is much improved, and my comments are mainly focused on grammatical error or awkward phrasing.

In general, the manuscript requires a review of comma usage. There are many instances where phrases are incomplete or not marked clearly by the presence of a comma, and the readability of this manuscript will improve with these edits. Similarly, there were few spelling errors that are noted in my line by line comments.

Lines 70-71: Add either "Due to a combination" or something along those lines to complete the phrase at the beginning of the sentence.

Added "Due to" in front of "a combination."

Line 72: I recommend using "we predict" rather than "we are predicted".

Agreed. Correction made.

Line 76: Not tolerant should be intolerant.

Correction made.

Line 83: comma after "tolerant as their prey"

Correction made.

Lines 102-104: This sentence reads awkwardly. "We expect cobia to handle warm conditions" is vague, and I respect that you don't want to go into too much detail here. Maybe use withstand

instead of handle. Additionally, the phrase after the semicolon is incomplete. "it is unlikely the intense predicted to occur in coastal habitats by the end-of-century."

Fixed this sentence and split into two separate sentences. "Cobia are known to frequent subtropical waters, therefore we expect cobia to have the capability to withstand warming waters over the next few decades. However, as conditions exponentially intensify by the end-of-century it is unclear whether they will be able to withstand those changes."

Line 108: comma between warm months and they are

Correction made.

Lines 114 and 115: possibly personal preference, but "caught and released cobia are released" reads poorly. Rephrase if possible.

Rephrased sentence to: "Cobia already have to endure capture, handling, and air exposure when caught and will have to face environmentally stressful conditions when released in the future"

Lines 121-123: How do repeated angling events on subsequent days indicate fast recovery time? I'd add a citation here.

Reworded and split into two sentences and added a citation indicating that feeding vs not feeding can help indicate presence of stress. "However, unpublished recreational catch and released data (S. Musick, pers. Comm.) show multiple cobia caught on subsequent days. This suggests stress levels have gone down enough where feeding is desired [39]; therefore, cobia may have a fast recovery time."

Line 156: Comma after experimental treatments to close off the introductory phrase

Correction made.

Line 224: Recommend changing "After being removed" to "After removal"

Correction made.

Line 252: Period before SMR.

Correction made.

Line 308: Comma at the end of the introductory phrase

Correction made.

Line 408: the crit subscript for Scrit is missing the t

Correction made.

Lines 416-417: The sentence mentioning how impressive this tolerance is creates subjectivity. Potentially remove or rephrase to emphasize the difference between cobia and other fishes without using "impressive"

Removed the sentence.

Line 448: I would remove the last sentence. Mentioning the limitations during your methods is sufficient.

Removed that sentence.

Lines 480-483: I appreciate the recommendations to fisheries, but I do not feel they need be so specific given the data collected in your paper. Instead, maybe suggest broader change, such as encouraging strategies to minimize hypoxia and thermal stress.

This comment contradicts the other reviewer's comment about expanding the section about fishing recommendations. We decided to follow the other reviewer's suggestions and expanded this section and made it a separate paragraph.

Figure 5 still suffers from readability issues. The dark coloration creates little contrast in the upper right hand corner.

We made the symbols bigger and put a lighter color for the border of the shapes to remove the contrast a little. The lack of contrast however is by design because if my model is predicting perfectly the background should match the colors on the shapes exactly. The lower the contrast between the shapes and the background, the better the model. We do understand discerning the shapes at all may be difficult for a colorblind person which was why we made the changes we did.